# A pore-forming protein drives macropinocytosis to facilitate toad water maintaining

Zhong Zhao [1,2,3,5], Zhi-Hong Shi [1,2,5], Chen-Jun Ye[1] & Yun Zhang [1,4✉]

Maintaining water balance is a real challenge for amphibians in terrestrial environments. Our previous studies with toad *Bombina maxima* discovered a pore-forming protein and trefoil factor complex βγ-CAT, which is assembled under tight regulation depending on environmental cues. Here we report an unexpected role for βγ-CAT in toad water maintaining. Deletion of toad skin secretions, in which βγ-CAT is a major component, increased animal mortality under hypertonic stress. βγ-CAT was constitutively expressed in toad osmoregulatory organs, which was inducible under the variation of osmotic conditions. The protein induced and participated in macropinocytosis in vivo and in vitro. During extracellular hyperosmosis, βγ-CAT stimulated macropinocytosis to facilitate water import and enhanced exosomes release, which simultaneously regulated aquaporins distribution. Collectively, these findings uncovered that besides membrane integrated aquaporin, a secretory pore-forming protein can facilitate toad water maintaining via macropinocytosis induction and exocytosis modulation, especially in responses to osmotic stress.

[1] Key Laboratory of Animal Models and Human Disease Mechanisms of Chinese Academy of Sciences/Engineering Laboratory of Peptides of Chinese Academy of Sciences, Kunming Institute of Zoology, Chinese Academy of Sciences, Kunming, Yunnan 650201, China. [2] Kunming College of Life Science, University of Chinese Academy of Sciences, Kunming, Yunnan 650204, China. [3] Institute of Basic Medical Sciences, Chinese Academy of Medical Sciences & Peking Union Medical College, Beijing 100005, China. [4] Center for Excellence in Animal Evolution and Genetics, Chinese Academy of Sciences, Kunming, Yunnan 650201, China. [5] These authors contributed equally: Zhong Zhao, Zhi-Hong Shi. ✉email: zhangy@mail.kiz.ac.cn

Maintaining water balance is a key challenge that amphibians face in the process of transition from water to land[1,2]. In these animals, the coordinated function of the nervous, endocrine and lymphatic systems and organs involved in water-salt balance and osmoregulation (including the skin, bladder and kidneys) results in enhanced water storage and utilization and reduced evaporation[1,3]. Amphibians are able to absorb water through their skin[1,2]. In addition, water is reabsorbed from the tubular fluid in the kidney and from stored urine in the urinary bladder (UB)[2,3]. Water channel proteins aquaporins (AQPs) play key roles in transepithelial water absorption/reabsorption in these organs, and in cell volume regulation[4,5]. However, it has rarely been studied on the physiological function of protein components from amphibian skin secretions in regulating of integumental water homeostasis.

Macropinocytosis is a mechanism that mediates bulk uptake and internalization of extracellular fluid and the solutes contained therein, producing endocytic vesicles with diameters of 0.2–5 μm[6,7]. Macropinocytosis is an actin-dependent endocytic pathway mediated by the activation of the Ras and phosphatidylinositol 3-kinase (PI3K)-signaling pathways, which is induced by either endogenous agents such as growth factors, or by invasive microbes[6,8]. This fundamental cellular process has been documented to play various patho-physiological roles in a range of normal and malignant cells, including nutrient acquisition, cell growth, traffic and renewal of membrane components, entry of pathogens, immune surveillance, and cellular motility[6,9,10]. However, the physiological roles of macropinocytosis, a very ancient form of endocytosis, remain incompletely understood[11,12].

Pore-forming proteins (PFPs) are usually secretory proteins that exist in a water-soluble monomeric form and oligomerize to form transmembrane pores (channels)[13,14]. Aerolysins are bacterial β-barrel PFPs produced by *Aeromonas* species[13,14]. Interestingly, numerous aerolysin family PFPs (abbreviated af-PFPs, previously referred as Aerolysin-Like Proteins, ALPs) harboring an aerolysin membrane insertion domain fused with other domains have been identified in various animals and plants[15,16]. Our previous studies with the skin secretions of the toad *Bombina maxima* discovered an interaction network among af-PFPs and trefoil factors (TFFs)[17]. BmALP1, an af-PFP from the toad, can be reversibly regulated between the active and inactive forms, in which its paralog BmALP3 is a negative regulator depending on environmental oxygen tension[18]. Specifically, BmALP1 interacts with BmTFF3 to form a membrane active PFP complex named βγ-CAT[19,20], in which BmTFF3 acts as an extracellular chaperon that stabilizes the BmALP1 monomer and delivers this PFP to its proper membrane targets[18,21].

This secretory PFP complex βγ-CAT targets gangliosides and sulfatides in cell membranes in a double-receptor binding model[21]. Then the BmALP1 subunit is endocytosed and this PFP oligomerizes to form channels on endolysosomes, modulating the contents and biochemical properties of these intracellular organelles[21–25]. Thus, βγ-CAT is a secretory endolysosome channel (SELC) protein. We have made a hypothesis that this PFP complex actually represents a hitherto unknown SELC pathway, which is characterized of mediating cellular material import and export through endolysosomal pathways[17]. Depending on cell contexts and surroundings, βγ-CAT have been proposed to promote the toad in the sense and uptake of environmental materials (like nutrients and antigens) and to facilitate the export of imported cargo molecules in their intact forms and/or processed products via stimulating extracellular vesicle releasing[17]. Thus, the cellular effects of SELC protein βγ-CAT could mediate material exchange between cells and environments, while maintaining mucosal barrier function and fulfilling immune defense[17]. Accordingly, the roles of βγ-CAT in immune defense have been first documented[22–26].

Amphibian skin is a major organ responsible for water acquisition and maintaining[1,3], and βγ-CAT is a major component of *B. maxima* skin secretions[18]. In the present study, we found that the expression and localization of βγ-CAT in toad *B. maxima* are related to environmental osmotic conditions, and that this protein is able to counteract cellular dehydration under extracellular hyperosmosis. βγ-CAT stimulated and participated in cell macropinocytosis and exosome release, promoting water and $Na^+$ uptake and regulating AQP localization. Collectively, these results revealed that a secretory PFP can drive water acquisition and maintaining.

## Results

**βγ-CAT is involved in responses to osmotic stress.** Amphibian skin has played a dual role in osmoregulation during evolutionary adaptation to land environments[27]. Accordingly, when exposed to hypertonic Ringer's solution, toads (*B. maxima*) rapidly lost weight, which was gradually recovered when the animal was then placed into isotonic Ringer's solution (Fig. 1a). This suggests that the toad transports water through its skin under conditions of osmotic stress. Skin secretions play pivotal roles in the physiological functions of amphibian skin[2]. Furthermore, βγ-CAT is a major proteinaceous component of skin secretions in toad *B. maxima*[18]. Two subunits of βγ-CAT together were estimated to account for about 50% of proteins in *B. maxima* skin secretions (Supplementary Fig. 1a). We speculated that toad skin secretions, especially βγ-CAT, might play an active role in water balance. Osmotic stress experiments were conducted to test the possible functions of βγ-CAT-containing skin secretions in maintenance of water balance in the toad. Toads *B. maxima* lost 20% of their body weight and 50% of the animals died when they were placed in the hypertonic Ringer's solution for 24 hours (Fig. 1b and Supplementary Fig. 1b). Toad skin secretions were depleted by electro-stimulation 30 minutes before the toads were placed in different osmotic solutions. Although no toad deaths were recorded after electro-stimulation in the isotonic Ringer's solution, mortality occurred in the hypertonic Ringer's solution (Fig. 1b). Therefore, it appears that skin secretions are critical for the toad to cope with hypertonic environments. To further investigate the possible involvement of βγ-CAT in the toad's water balance, qPCR was used to detect the expression of two βγ-CAT subunits in the skin, UB and kidney of toad *B. maxima* during dehydration and weight recovery (water absorption after dehydration). It was found that *βγ-CAT α*-subunit expression was upregulated in the skin and kidney during weight recovery via water uptake after dehydration (Fig. 1c and Supplementary Fig. 1c). In contrast, the expression of *βγ-CAT β*-subunit mRNA was upregulated in the toad skin during dehydration or water absorption (Fig. 1d), but not in the kidney (Supplementary Fig. 1d). Interestingly, in the UB, the mRNA levels of the both subunits were increased when toads were exposed to hypertonic Ringer's solution, and returned to the normal level when toads were subsequently placed in isotonic Ringer's solution (Fig. 1e, f). Therefore, βγ-CAT expression is associated with changes to external osmotic conditions.

In addition, we analyzed the expression and localization of βγ-CAT protein under the same treatment conditions via immuno-histofluorescence (IHF). βγ-CAT was mainly concentrated within the epidermis and glands of the skin, and it was upregulated in the epidermis and basement membrane during osmotic stress (Fig. 1g). βγ-CAT was predominantly localized within the transitional epithelium in the UB. The expression of βγ-CAT protein in toad UB was upregulated in hypertonic Ringer's solution and downregulated when the toad was then placed into isotonic Ringer's solution (Fig. 1h). Collectively, these results revealed an involvement of βγ-CAT in the toad's responses to osmotic stress.

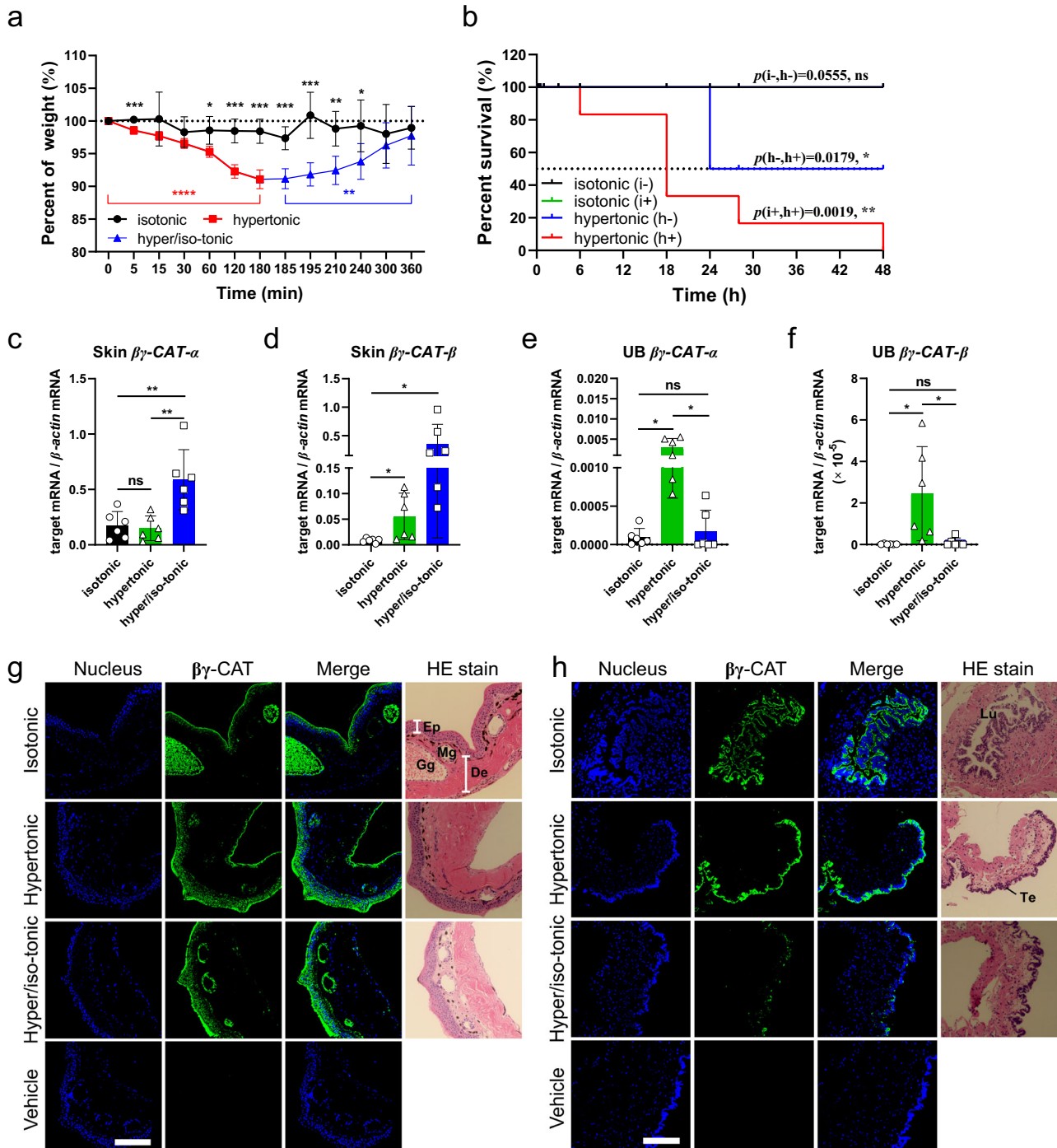

**Fig. 1 βγ-CAT is involved in responses to osmotic stress. a** Toad weight changes were measured after placing them in isotonic, hypertonic and hypertonic/isotonic Ringer's solutions. Initial weight of toads is 19 ± 5 g ($n = 5$). **b** Survival rates of toads in isotonic (i) and hypertonic (h) Ringer's solution were determined after 48 hours. Toads were placed in each of the solutions after 30 minutes with (+) or without (-) electro-stimulation to delete toad skin secretions ($n = 6$). **c-f** The expression of *βγ-CAT* subunits in toad skin and UB was analyzed by real-time fluorescent quantitative PCR ($n = 6$). Toads were placed in isotonic or hypertonic Ringer's solution for 3 hours before the samples were collected. In the hypertonic/isotonic group, the toads were first placed in the hypertonic solution for 3 hours, then moved to isotonic solution for a further 3 hours before the samples were collected. **g, h** Following placement of toads in isotonic, hypertonic or hypertonic/isotonic Ringer's solution as described above, the localization and expression of βγ-CAT in the toad skin (**g**) and UB (**h**) tissues were analyzed by immunohistofluorescence (IHF). Ep epidermis, De dermis, Mg mucous gland, Gg granular gland, Lu lumen, Te transitional epithelium. Scale bars, 200 μm. *$P < 0.05$ and **$P < 0.01$ by the Gehan-Breslow-Wilcoxon test in survival rate analysis. ns ($P \geq 0.05$); *$P < 0.05$, **$P < 0.01$ and ***$P < 0.001$ by unpaired *t* test in other experiments. All data represent the mean ± SD and are representative of at least two independent experiments. See also supplementary Fig. 1.

**βγ-CAT counteracts cell dehydration under extracellular hyperosmosis.** βγ-CAT is a vital functional protein of toad *B. maxima*, and its constitutive expression can be detected in various toad tissues[22,23], including skin secretions and epithelial cells from the skin and UB, as well as kidney and peritoneal cells. Endogenous secretion of βγ-CAT in these toad-derived cells was further analyzed by a hemolysis assay, a sensitive method to detect the presence of biologically active βγ-CAT[18,19]. All the media of these cultured toad cells showed potent hemolytic activity, which was totally inhibited by anti-βγ-CAT antibodies, confirming the presence of secreted βγ-CAT (Supplementary Fig. 2a). In addition, we measured the cytotoxicity of βγ-CAT to *B. maxima* cells. Previously, it was reported that βγ-CAT showed no cytotoxicity to toad peritoneal cells at dosages up to 400 nM[22]. The present study further determined that cytotoxicity of βγ-CAT to skin and UB epithelial cells and kidney cells of *B. maxima* occurred only when its concentration reached 2 μM (Supplementary Fig. 2b), a level much higher than physiological concentrations (20–100 nM, as determined in the toad peritoneum)[22]. In contrast, mammalian cells are much more sensitive to βγ-CAT. Although βγ-CAT showed no cytotoxicity to MDCK, Caco-2 and T24 cells at 10 nM, the protein caused cell death when dosages used were higher than 50–100 nM (Supplementary Fig. 2c). Thus, in all subsequent experiments concerning the addition of purified βγ-CAT to cells, the protein dosages used were: epithelial cells from toad skin, 100 nM; UB epithelial cells, 50 nM; peritoneal cells, 50 nM; mammalian MDCK, 10 nM; Caco-2, 10 nM; and T24, 5 nM.

To further explore the potential role of βγ-CAT in water transport, we investigated whether the protein is able to counteract cell dehydration under extracellular hyperosmosis. We confirmed the action of βγ-CAT on toad UB epithelial cells, mammalian MDCK, Caco-2 and T24 cells, as determined by the formation of βγ-CAT oligomers after treatment with the protein (Fig. 2a and Supplementary Fig. 2d). βγ-CAT oligomers were detected in toad UB epithelial cells without the addition of the protein, possibly caused by endogenously secreted βγ-CAT (Fig. 2a). The addition of purified βγ-CAT further increased the concentration of oligomers of the protein in the UB epithelial cells (Fig. 2a). Then, we analyzed the electrophysiological characteristics of βγ-CAT oligomers on outside-out patches of HEK293 cells. βγ-CAT induced macroscopic currents were recorded and normalized, which has the characteristic of inward rectification (Fig. 2b). These proved that βγ-CAT could form transmembrane pores (channels) and mediate ion flow after oligomerization on the membrane. We further investigated ion selectivity of βγ-CAT channels through ion replacement experiments. Asymmetric 150:15 mM NaCl solutions left-shifted the reversal potential from 0 mV to −61.2 mV, which is very close to the theoretical equilibrium potential of $Na^+$, indicating that the βγ-CAT channels are permeable to $Na^+$ but not to $Cl^-$ (Fig. 2c). These results demonstrated that the βγ-CAT channels allowed $Na^+$ flow.

On this basis, we explored the potential role of βγ-CAT when cells were challenged by extracellular hyperosmosis using hypertonic solution. We observed that βγ-CAT promoted volume recovery of MDCK, Caco-2 and T24 cells in hypertonic solution (Fig. 2d), revealing the capacity of the protein to facilitate water uptake under hyperosmotic conditions. In consistency with those observations, immune-depletion of endogenous βγ-CAT further decreased the cell diameters of toad UB epithelial cells in hypertonic Ringer's solution relative to those in the absence of anti-βγ-CAT antibodies (Fig. 2e). It is well documented that AQPs can mediate the rapid cellular flow of water[28]. Thus, it is necessary to clarify whether the cell volume recovery by water uptake mediated by βγ-CAT under osmotic stress was associated with AQP function. Because the mercury-sensitive sites of *B.*

*maxima* AQPs (BmAQPs) in the UB were evolutionarily conserved (Supplementary Fig. 2e, f), $HgCl_2$ was used to inhibit BmAQPs function. Cell volume was unchanged in hypertonic Ringer's solution after blocking BmAQPs with $HgCl_2$. Conversely, cell volume recovery of toad UB epithelial cells via water uptake was clearly reduced after immunodepletion of endogenous βγ-CAT (Fig. 2f). Taken together, these results revealed the capacity of βγ-CAT to promote cell volume recovery by stimulating water acquisition during extracellular hyperosmosis, and indicated that the effect was independent of water flow via plasma membrane AQPs.

**βγ-CAT promotes macropinocytosis.** βγ-CAT participates in cell volume regulation, suggesting its involvement in water transport. We next analyzed the possible cellular mechanism behind this phenomenon. Water acquisition can be rapidly realized through AQPs and/or macropinocytosis[12,28,29]. Previously, βγ-CAT enhanced pinocytosis in a murine dendritic cell (DC) model[24]. In the present study, we carefully studied the ability of βγ-CAT to stimulate and participate in macropinocytosis in various types of *B. maxima* cells. First, immunoelectron microscopy (IEM) revealed that βγ-CAT was localized in cellular pseudopodia and macropinosomes with diameters up to 300 nm formed by macropinocytosis in toad skin and UB tissues (Fig. 3a). βγ-CAT was also present in the intercellular spaces of epithelial cells (Fig. 3a). 70-kDa dextran is a marker for macropinocytosis, while Lucifer Yellow (LY) can be used as a marker of fluid-phase endocytosis including macropinocytosis[30,31]. In epithelial cells obtained from toad skin and UB, and in toad kidney or peritoneal cells, immunodepletion of endogenous βγ-CAT by anti-βγ-CAT antibodies greatly decreased internalization of LY and FITC-dextran, relative to cells observed in the presence of control rabbit IgG (Fig. 3b, c and Supplementary Fig. 3a, b). Surprisingly, exogenously added control rabbit IgG can induce pinocytosis. The widespread presence of IgG receptors (e.g., FcRn) in cells may contribute part of the effect[32]. In fact, the *FcRn*-like gene of *B. maxima* was also present in toad cells (Supplementary Fig. 3c). For unknown reasons, rabbit IgG contributes to pinocytosis for toad cells as an exogenous protein. We found that control rabbit IgG and anti-βγ-CAT rabbit-derived antibodies were equally potent in inducing macropinocytosis in MDCK cells (Supplementary Fig. 3d). Therefore, these demonstrated that anti-βγ-CAT rabbit-derived antibodies can reduce βγ-CAT-induced macropinocytosis by immunodepletion of endogenous βγ-CAT. Furthermore, addition of βγ-CAT to cells from toad skin and UB, and mammalian MDCK and T24 cells enhanced the internalization of LY and FITC-dextran (Fig. 3d, e and Supplementary Fig. 3e, f). We found that βγ-CAT was endocytosed and located in the macropinosomes of MDCK cells (Supplementary Fig. 3g). In addition, total $Na^+$ concentrations in toad UB epithelial cells and MDCK cells treatment with βγ-CAT were 3.5 and 1.1 times of that of the vehicle group, respectively (Fig. 3f).

Next, the molecular mechanism of macropinocytosis induced by βγ-CAT was investigated using pharmacological agents. 5-(N-ethyl-N-isopropyl) amiloride (EIPA, a $Na^+/H^+$-exchanger inhibitor) and wortmannin (WORT, a PI3K inhibitor) are commonly used to inhibit macropinocytosis[33,34]. We found that EIPA and WORT reduced macropinocytosis induced by βγ-CAT in toad UB epithelial cells and mammalian MDCK cells (Fig. 3g, h). Moreover, βγ-CAT stimulated the phosphorylation of Akt and PI3K and the expression of Rac123 in MDCK cells (Fig. 3i). Taken together, these results demonstrated that βγ-CAT, an endogenously secreted complex of a PFP and a TFF, stimulates and participates in macropinocytosis in diverse cells of *B. maxima* as well as in various mammalian cells.

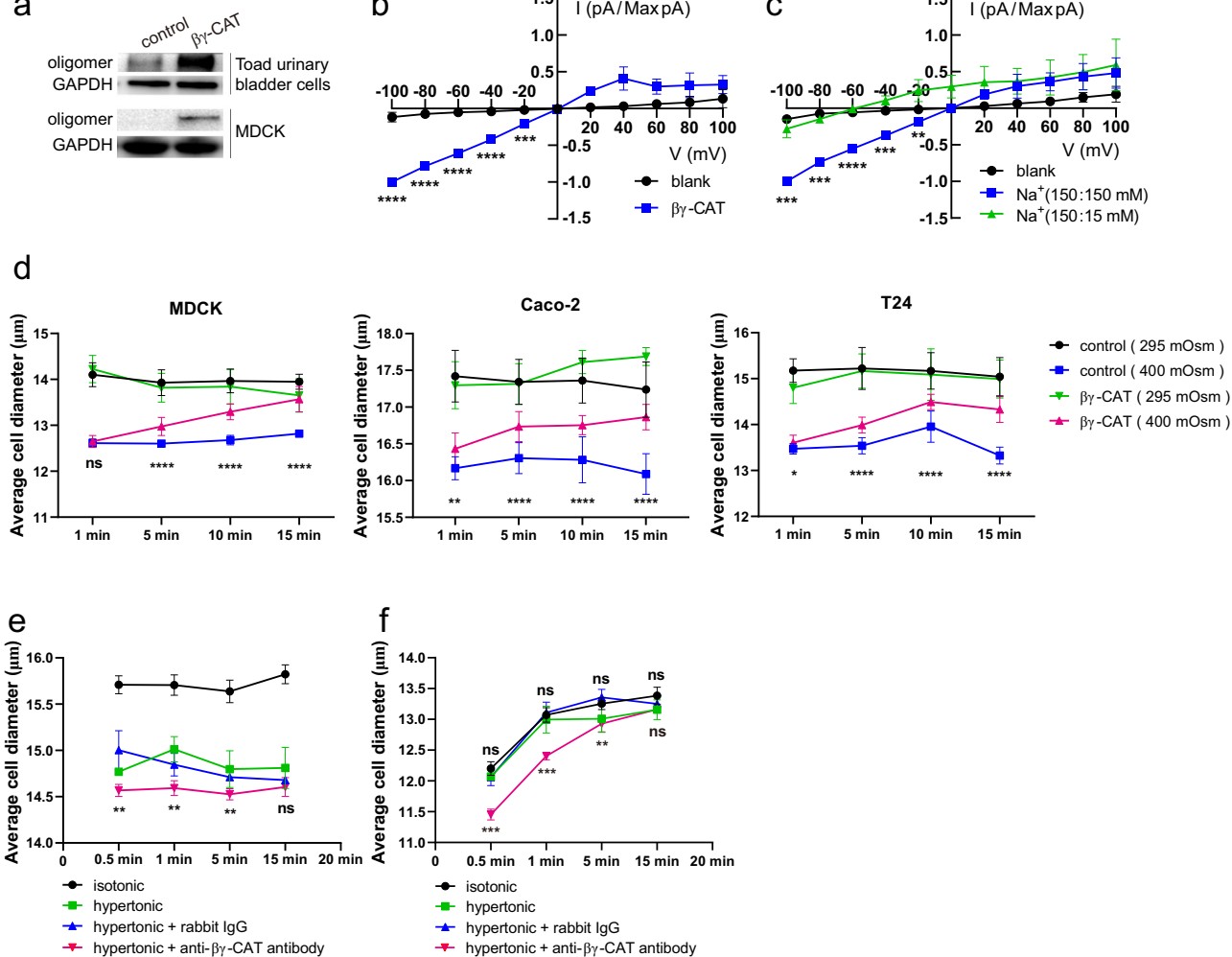

**Fig. 2 βγ-CAT counteracts cellular dehydration under extracellular hyperosmosis. a** Toad urinary bladder cells and MDCK cells were treated with or without βγ-CAT for 15 minutes. The appearance of βγ-CAT oligomers in the treated cells was determined by Western blotting. **b** Normalized current-voltage curves of channels formed by 100 nM βγ-CAT on HEK293 cells. Currents were elicited by 500 ms ramp protocol between −100 mV to +100 mV every 2 seconds from a holding potential of 0 mV ($n = 3$). **c** Normalized current-voltage curves of βγ-CAT channels in indicated solutions (pipette/bath) ($n = 3$). Na$^+$(150:15 mM) represents the 150 mM Na$^+$ in the pipette and 15 mM Na$^+$ in the bath. **d** Diameter changes of MDCK, Caco-2 and T24 cells in isotonic (295 mOsm) or hypertonic (400 mOsm) PBS in the presence or absence of βγ-CAT as determined by using a cell counter. Digested cells were first suspended in PBS for 15 minutes before they were used in this experiment. **e, f** Diameter changes of toad UB epithelial cells in isotonic (black) and hypertonic (green) Ringer's solution in the presence of 50 μg mL$^{-1}$ rabbit antibody (blue) or anti-βγ-CAT antibody (magenta). In (**f**), the cells were first treated with 0.3 mM HgCl$_2$ for 10 minutes before cell diameter measurement with a cell counter. In all experiments shown in Fig. 2, βγ-CAT dosages used were 10 nM for MDCK and Caco-2, 5 nM for T24 and 50 nM for toad UB epithelial cells. The bars represent the mean ± SD of triplicate samples in **b–e**. The bars represent the mean ± SD of three independent replicates in **f**. ns ($P ≥ 0.05$), *$P < 0.05$, **$P < 0.01$, ***$P < 0.001$ and ****$P < 0.0001$ by the unpaired $t$ test. All data are representative of at least two independent experiments. See also supplementary Fig. 2.

**βγ-CAT in AQP regulation.** AQPs and ion flux are involved in water transport and volume regulation of toad UB cells under stimulation by multiple hormones[35–37]. Internalization by macropinocytosis is unselective, whereas transitional epithelial cells are polar. Therefore, we assessed the relationship between βγ-CAT-stimulated macropinocytosis and BmAQP2. We observed that the location of βγ-CAT and BmAQP2 in toad UB epithelial cells differed between isotonic, hypertonic and 'hypertonic/isotonic' (hypertonic followed by returning to isotonic) Ringer's solutions (Fig. 4a). When toads were exposed to isotonic Ringer's solution, βγ-CAT and BmAQP2 were located together and were distributed on both the apical and basal sides of the transitional epithelium in toad UB. In hypertonic Ringer's solution, βγ-CAT and BmAQP2 shifted their location to the basal or lateral sides of the UB transitional epithelium. In contrast, when the toads were

returned to isotonic Ringer's solution from hypertonic Ringer's solution, βγ-CAT and BmAQP2 began to migrate to the apical side of the transitional epithelium. This observation suggests that macropinocytosis induced by βγ-CAT is involved in the internalization and transport of BmAQP2 in the UB. In MDCK, intracellular colocalization of βγ-CAT and AQP2 was also observed (Fig. 4b). These results indicated that macropinocytosis stimulated by βγ-CAT plays a role in the regulation of AQPs under various osmotic conditions, revealing another aspect of this protein in the modulation of toad water balance.

**βγ-CAT enhances exosome release.** The secretion of exosomes is an important aspect of cell exocytosis that can be viewed as a means of selective transport of materials among cells and a mode of intercellular communications[38]. Because βγ-CAT promotes

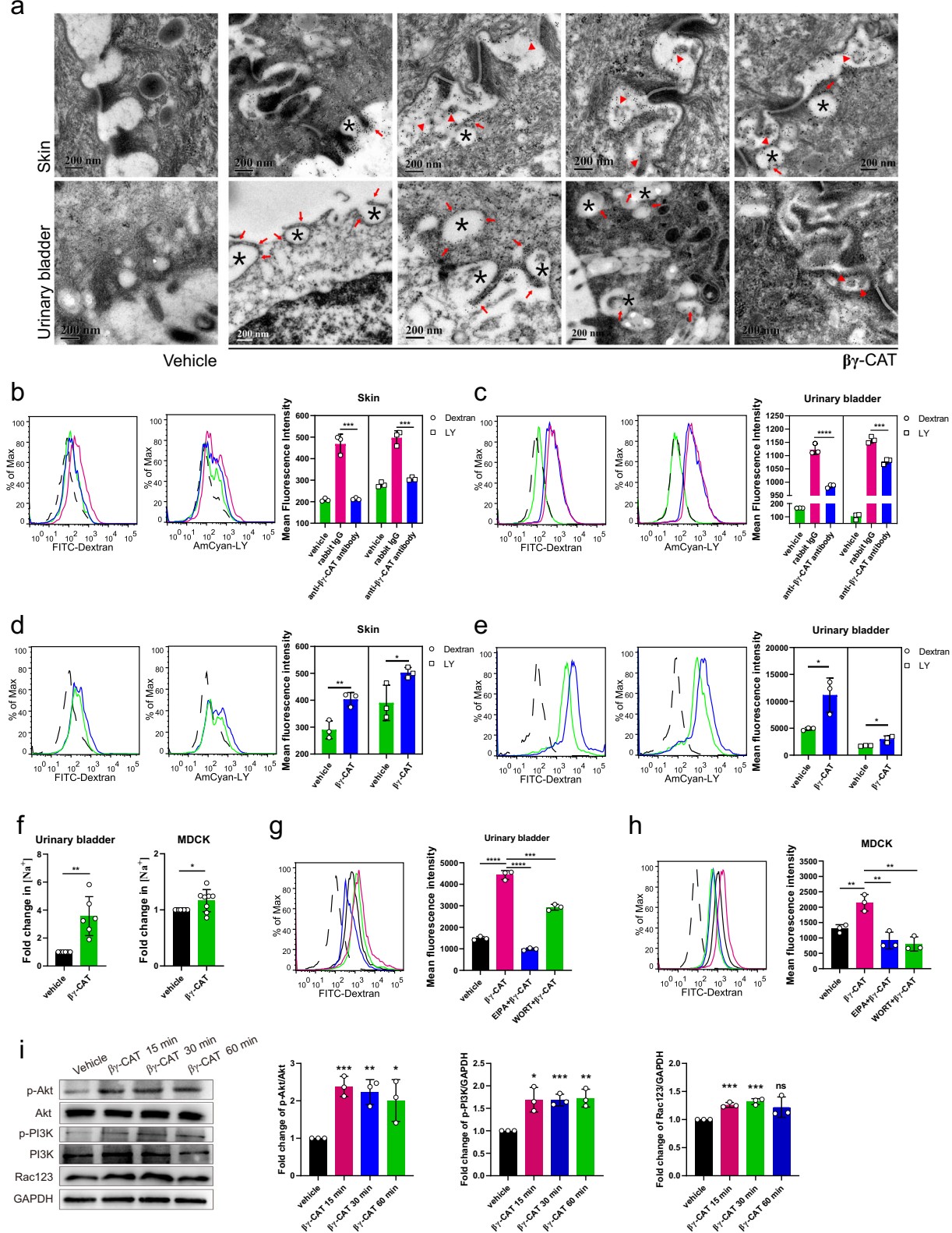

macropinocytosis, we further explored the cellular fate and role of βγ-CAT-containing vesicles. βγ-CAT was detected in both multivesicular bodies (MVBs) and intraluminal vesicles (ILVs) of UB epithelial cells (Fig. 5a). Considering that 98% of toad UB epithelial cells survived 3 hours in vitro culture at room temperature (Supplementary Fig. 4a), exosomes secreted by these cells were collected and observed (Fig. 5b), which contained typical exosome

markers CD63, TSG101 and flotillin-1, and the number of exosomes was increased when these cells were treated with purified βγ-CAT (Fig. 5c). Nanoparticle tracking analysis (NTA) revealed that immunodepletion of endogenous βγ-CAT attenuated exosome release from toad UB epithelial cells and peritoneal cells (Fig. 5d and Supplementary Fig. 4b), while the addition of βγ-CAT substantially augmented exosome release from these cells

**Fig. 3 βγ-CAT promotes macropinocytosis. a** Ultrastructural localization of βγ-CAT in toad skin and UB tissues as analyzed by IEM. Vehicle (rabbit IgG control). Endocytic vesicles formed by macropinocytosis (black asterisks), and distribution of βγ-CAT on vesicles (red arrows) and intercellular spaces (red triangles). **b, c** Immunodepletion of endogenous βγ-CAT decreased macropinocytosis. Toad skin (**b**) and UB (**c**) epithelial cells were incubated with 50 μg mL$^{-1}$ anti-βγ-CAT antibody to immunodeplete endogenous βγ-CAT for 30 min. The mean fluorescence intensity was determined by flow cytometry with 100 μg mL$^{-1}$ of 70 kDa FITC-label dextran and Lucifer Yellow (LY) for 30 minutes. Rabbit IgG (antibody control). Vehicle (antibody absent control). **d, e** The addition of purified βγ-CAT augmented macropinocytosis. The mean fluorescence intensity of LY and FITC-label dextran in toad skin (**d**) and UB (**e**) epithelial cells was determined by flow cytometry with or without additional 100 nM or 50 nM βγ-CAT, respectively. **f** Fold changes of [Na$^+$] in toad UB epithelial cells (n = 6) and MDCK cells (n = 8) with and without the addition of 50 nM or 10 nM βγ-CAT for 3 hours, respectively. **g, h** The effect of inhibitors on macropinocytosis induced by βγ-CAT. Toad UB epithelial cells (**g**) and MDCK cells (**h**) were incubated with and without 100 μM EIPA or 20 μM WORT for 1 hour. Then the cells were cultured with 100 μg mL$^{-1}$ FITC-label dextran with 50 nM (toad UB cells) or 10 nM (MDCK cells) βγ-CAT for 30 minutes. **i** Rac123 and phosphorylation of Akt and PI3K in response to 10 nM βγ-CAT in MDCK cells for 15, 30 or 60 minutes as determined by Western blotting and bands were semiquantified with ImageJ. The black dotted line refers to the blank control, and data represent the mean ± SD of triplicate samples in **b-e, g, h**. Data represent the mean ± SD of at least three independent replicates in **f** and **i**. ns (P ≥ 0.05), *P < 0.05, **P < 0.01, ***P < 0.001 and ****P < 0.0001 by the unpaired t test. All data are representative of at least two independent experiments. See also supplementary Fig. 3.

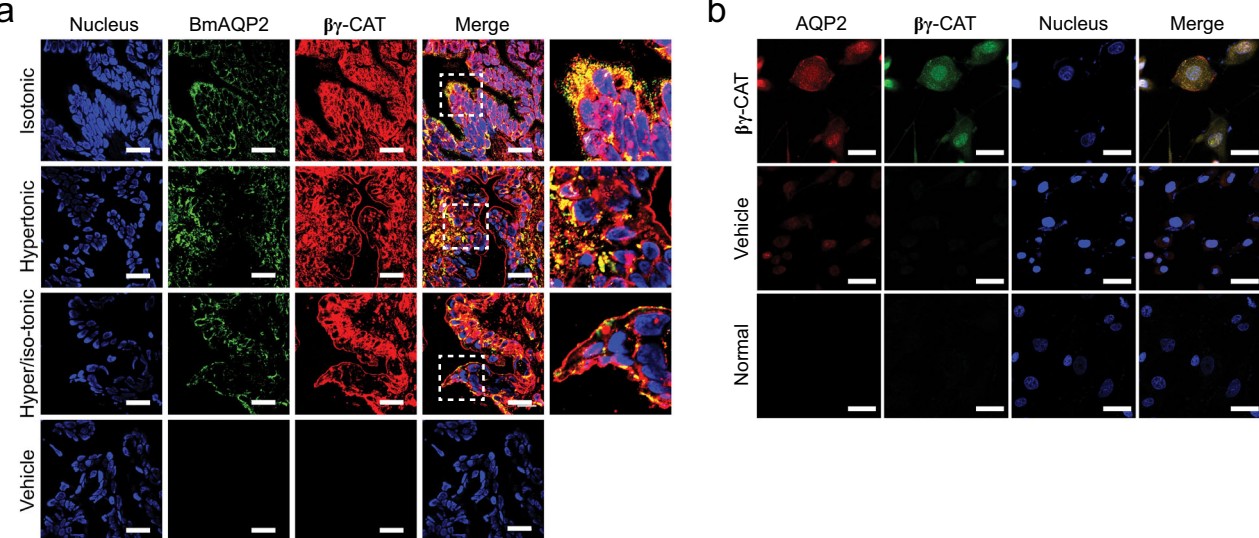

**Fig. 4 βγ-CAT in AQP regulation. a** Colocalization of βγ-CAT and BmAQP2 in the UB tissue of toad *B. maxima* after toads were placed in isotonic, hypertonic or hypertonic/isotonic Ringer's solution for 3 hours as analyzed by immunohistofluorescence. Scale bars, 25 μm. **b** Intracellular colocalization of βγ-CAT and AQP2 in MDCK cells with or without the treatment of 10 nM βγ-CAT for 15 minutes as determined by immunofluorescence. Scale bars, 30 μm. All data are representative of at least two independent experiments.

(Fig. 5e and Supplementary Fig. 4c). None of these treatments changed the average exosome diameter. Similar results were also obtained in mammalian MDCK and T24 cells (Supplementary Fig. 4d, e). Collectively, these results showed that βγ-CAT promotes the production and release of exosomes in both toad and mammalian cells.

We further investigated the properties of exosomes stimulated by βγ-CAT. The size distribution of exosomes derived from toad UB epithelial cells was determined by flow cytometry for nanoparticle analysis. The diameters of these exosomes were mainly concentrated in the range of 50–150 nm (Supplementary Fig. 4f). When 1 mg mL$^{-1}$ of FITC-dextran was incubated with isolated exosomes, the exosomes did not contain FITC-dextran (Supplementary Fig. 4g). In contrast, when dextran was added to the culture of toad UB epithelial cells under the same conditions, FITC-dextran was identified in exosomes secreted by the cells (Supplementary Fig. 4g). This observation indicated that isolated exosomes did not take up 70-kDa dextran, but dextran was taken up by toad UB epithelial cells by macropinocytosis and released from cells in the form of exosomes. Interestingly, the proportion of exosomes containing FITC-dextran remained unchanged in toad UB epithelial cells with or without the addition of purified βγ-CAT (Fig. 5f), and there was no difference in the mean fluorescence

intensity of these exosomes (Fig. 5g). These results suggested that βγ-CAT promotes exocytosis and facilitates the transcellular transport of extracellular substances like dextran by increasing exosome release. However, the apparent properties of exosomes released seem not altered as assayed at the present stage.

Both BmAQP2 and βγ-CAT oligomers were detected by Western blotting in exosomes released from toad UB epithelial cells, reflecting the presence of endogenous βγ-CAT. The addition of βγ-CAT to cells greatly enhanced the quantity of BmAQP2 and βγ-CAT oligomers detected (Fig. 5h). IEM demonstrated colocalization of BmAQP2 and βγ-CAT in individual exosomes (Fig. 5i). These observations suggested that βγ-CAT drives the extracellular recycling and/or intercellular communication of AQPs via exosome release.

Furthermore, we analyzed the effect of βγ-CAT on Na$^+$ levels in exosomes from MDCK cells. No difference was observed for total Na$^+$ concentrations in exosomes produced by the same number of cells treated with or without 10 nM βγ-CAT (Fig. 5j). However, βγ-CAT increased exosome release of MDCK by 6.8 times (Supplementary Fig. 4d), so the mean Na$^+$ concentrations of exosomes were reduced (Fig. 5k). In addition, an increase in total Na$^+$ concentrations in exosomes induced by βγ-CAT were found in hypertonic media under the same conditions (Fig. 5l).

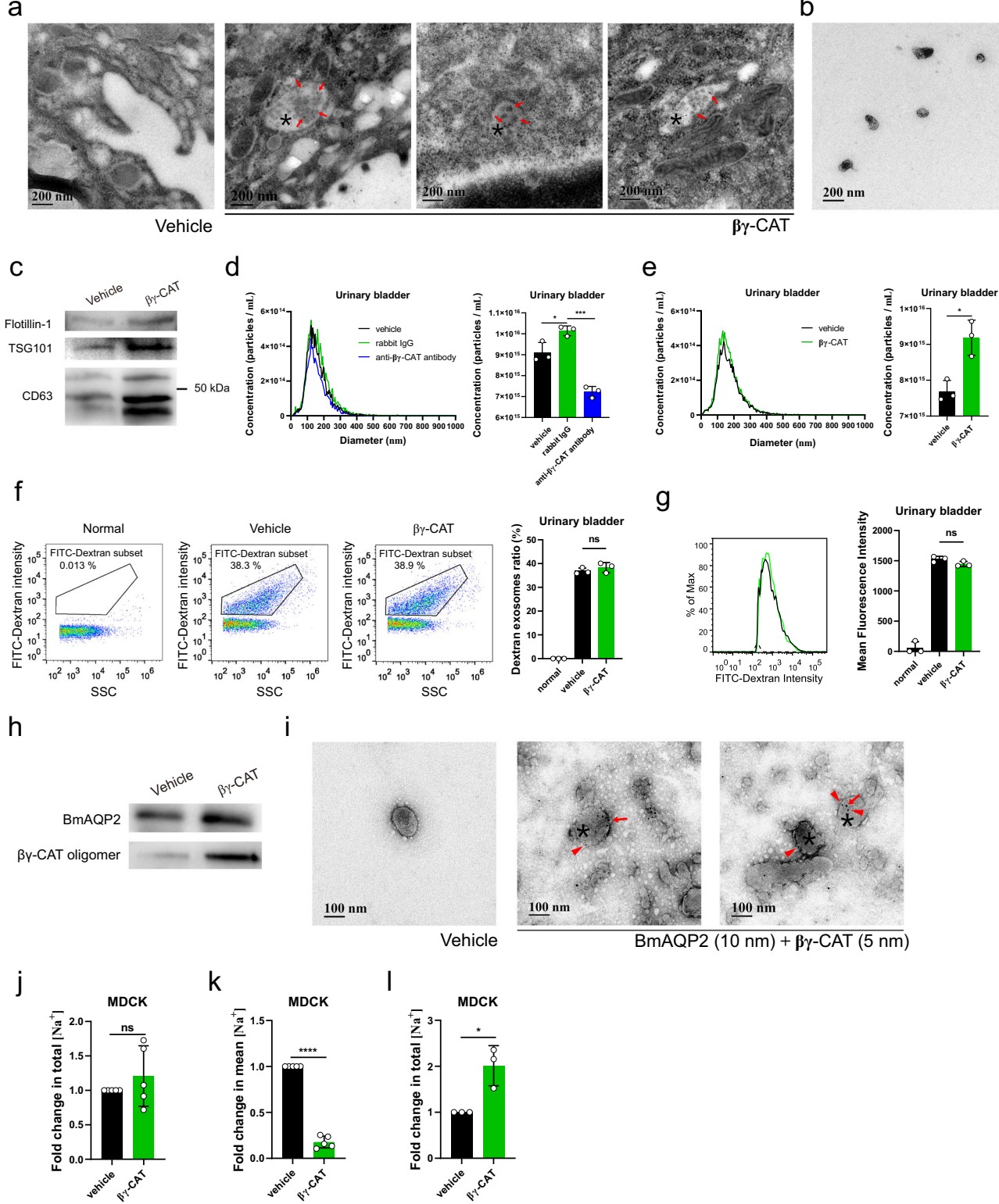

## Discussion

Amphibians live both on land and in water, and their skin is naked and acts as an important organ in water balance, respiration (gas exchange) and immune defense[1,3]. Though initially identified in *B. maxima* skin secretions[19], the PFP complex βγ-CAT is widely expressed in *B. maxima* water balance organs, including skin, UB and kidney (Fig. 1c–h, and Supplementary Fig. 1c, 1d). Previous studies indicated the expression of βγ-CAT in various *B. maxima* tissues[22,23]. Interestingly, in contrast to its potent cytotoxicity to various mammalian cells[19,39], βγ-CAT only shows toxic effects in toad cells in a dosage up to 2 μM (Supplementary Fig. 2b), which is much higher than βγ-CAT physiological concentrations observed[22]. Its toxicity to mammalian cells could reflect the absence of regulatory mechanisms for this

**Fig. 5 βγ-CAT enhances exosome release. a** Ultrastructural localization of βγ-CAT in toad UB tissue by IEM. βγ-CAT (arrow) was readily detected in MVBs (asterisk) or ILVs. Vehicle (rabbit IgG control). **b** TEM analysis of exosomes from the supernatant of toad UB epithelial cells cultured in vitro for 3 hours. **c** Western blotting analysis of flotillin-1, TSG101 and CD63 molecules in exosomes isolated from the supernatant of toad UB epithelial cells cultured in vitro for 3 hours with or without the addition of 50 nM βγ-CAT. **d, e** Analysis of the concentration and particle size (30–200 nm) of exosomes from toad UB epithelial cells with and without 50 μg mL$^{-1}$ anti-βγ-CAT antibodies (**d**) or the addition of 50 nM βγ-CAT (**e**) by NTA. **f, g** The percentage (**f**) and mean fluorescence intensity (**g**) of dextran-containing exosomes from toad UB epithelial cells in a culture medium containing 1 mg mL$^{-1}$ FITC-label dextran by Nanoflow Cytometry with or without the addition of 50 nM βγ-CAT. **h** Western blotting analysis of βγ-CAT and BmAQP2 in exosomes of toad UB epithelial cells cultured in vitro for 3 hours with or without the addition of 50 nM βγ-CAT. **i** IEM determination of βγ-CAT and BmAQP2 in exosomes (asterisk) from toad UB epithelial cells. BmAQP2 and βγ-CAT were labeled with 10-nm (arrow) and 5-nm (triangle) colloidal gold particles, respectively. **j, k** Fold changes of total (**j**) and mean (**k**) Na$^+$ concentrations in exosomes from the same number of MDCK cells with or without 10 nM βγ-CAT for 3 hours. **l** Fold changes of total Na$^+$ concentrations in exosomes from the same number of MDCK cells under the hypertonic medium with or without 10 nM βγ-CAT for 3 hours. The bars represent the mean ± SD of triplicate samples in **d–g**. Data represent the mean ± SD of at least three independent replicates in **j–l**. ns ($P \geq 0.05$), *$P < 0.05$ and ****$P < 0.0001$ by the unpaired $t$ test. All data are representative of at least two independent experiments. See also supplementary Fig. 4.

---

exogenous PFP protein, which are normally present in the toad[18,21]. These observations support the notion that the PFP complex βγ-CAT can not be simply viewed as a cell death inducer and/or a microbicide, and besides its roles in immune defense[22–26], the protein exhibits other essential physiological functions in this toad.

*B. maxima* skin is covered with skin secretions containing various biological molecules that fulfill a range of physiological function, including hormone-like peptides, antimicrobial peptides, af-PFPs, TFFs, and haem b-containing albumin[16,40,41]. Indeed, our results revealed that the skin secretions are necessary to prevent dehydration (Fig. 1b). We further demonstrated that a major skin secretion component βγ-CAT has a role in toad water homeostasis in osmoregulatory organs. Although most amphibians do not live in hypertonic environments and their skin is rarely challenged by extreme water loss, the kidney and UB are subject to a variety of hypertonic challenges under physiological conditions such as urine formation and concentration. In fact, the differential expression of βγ-CAT in UB with osmotic environment suggests that βγ-CAT is involved in the dynamic change of UB water transport (Fig. 1e, f and Fig. 4a). Therefore, we further demonstrated that βγ-CAT has a role in toad water homeostasis with toad UB cells and mammalian kidney cells. This PFP complex possesses the capacity to stimulate and participate in macropinocytosis (import) and exosome release (export) by epithelial cells of toad osmoregulatory organs, which promotes water and Na$^+$ transport involved in water homeostasis. It should be emphasized that since *B. maxima* is a non-model species, it is currently difficult to further explore the physiological effects of βγ-CAT at the animal level by means of gene knockout.

Previous studies with mammalian cells and toad peritoneal cells showed that βγ-CAT exerts its biological actions by endocytosis and the formation of channels on endolysosomes[17,21–24], but the endocytic pathway of the protein remains elusive. Macropinocytosis, also referred to as cell drinking, is a form of endocytosis that mediates the non-selective uptake of extracellular fluid and solutes[6,12]. In a murine DC model, βγ-CAT increases the internalization of ovalbumin (antigen), presumed to be mediated by enhanced macropinocytosis[24]. The present study carried out on various toad-derived cells, including epithelial cells from osmoregulatory organs and peritoneal cells, clearly showed that βγ-CAT is capable of inducing and participating in macropinocytosis. Macropinocytosis stimulated by βγ-CAT has been shown to facilitate the entry of water and Na$^+$ into cells as assayed in epithelial cells from toad osmoregulatory organs (Fig. 2f and Fig. 3f). This may explain the role of βγ-CAT in promoting the recovery of cell volume under hypertonic stress (Fig. 2d–f). Given that IgG binds widely distributed IgG Fc receptors to induce pinocytosis[32], such as MHC-class-I-like IgG

receptor protein FcRn, the co-presence of non-specific exogenous rabbit IgG and active βγ-CAT may lead to an increase in macropinocytosis (Fig. 3b, c and Supplementary Fig. 3a, b). Growth factor-induced macropinocytosis is mediated by the activation of the Ras and PI3-kinase signaling pathways[11]. Similarly, activation of PI3-kinase and Akt signaling was involved in macropinocytosis stimulated by βγ-CAT, as suggested by the effects of pharmacological inhibitors and the phosphorylation analysis of PI3K and Akt (Fig. 3g–i). However, in contrast to classic growth factors, which bind to membrane protein receptors to initiate their signaling, βγ-CAT targets gangliosides and sulfatides as receptors in lipid rafts to initiate its cellular effects[21]. The signals downstream from these lipid components and their relationship to the induction of macropinocytosis are presently unclear. In this respect, possible differences between macropinocytosis stimulated by growth factors and that induced by βγ-CAT should be investigated.

While engulfing large volumes of fluid, macropinocytosis also internalizes cell surface proteins such as receptors and integrins[42,43]. It has been proposed that endocytosis stimulated by βγ-CAT plays a role in the sorting of specific plasma membrane elements, such as functional integrated proteins or lipid components, which regulate cell responses to environmental variations[17]. In addition, internalization and recycling of AQPs between the plasma membrane and the endosomal compartment have roles in controlling water uptake and conservation[5,28]. Accordingly, colocalization of βγ-CAT with AQPs was observed in endocytic organelles (Fig. 4), suggesting macropinocytosis induced by this PFP protein drives endocytosis and recycling of AQPs. Clearly, the presence of AQPs in the plasma membrane might facilitate water loss when cells faced by hypertonic osmotic stress, thus internalization of AQPs induced by βγ-CAT could counteract dehydration by preventing rapid water efflux and cell shrinkage.

The secretion of extracellular vesicles comprising exosomes and microvesicles represents a novel mode of intercellular communication and material exchange[38,44]. Previously, βγ-CAT was found to augment βγ-CAT-containing exosome release from murine DCs, which activates T cell response effectively[24]. Because βγ-CAT is a factor exogenous to murine DCs, the explanation of the phenomenon is not obvious. However, the PFP is an endogenous element to toad cells. The present study illustrated that cell exocytosis in the form of exosome release was indeed augmented in the presence of βγ-CAT, as determined in diverse toad-derived cells (Fig. 5d, e and Supplementary Fig. 4b, c), revealing that mediation of cell exocytosis via exosome release is an intrinsic property of this PFP protein. The FcRn expressed by many cells, which can promote exocytosis and recycling of IgG[45]. The exocytosis effect of βγ-CAT may be further amplified by

exocytosis and circulation of IgG (Fig. 5d and Supplementary Fig. 4b). Previous studies demonstrated that βγ-CAT characteristically neutralizes the acidification of endocytic organelles containing this protein[22–24]. This cellular process may result in the transformation of βγ-CAT-containing endocytic organelles into MVBs that does not fuse with lysosomes for the degradation of contained solutes. The presence of extracellular dextran as a βγ-CAT-induced tracer of endocytosis in βγ-CAT-containing exosomes also supports the above view (Fig. 5f, g).

The observation that βγ-CAT both stimulates and participates in macropinocytosis and exosome release strongly implies that this PFP is involved in transcellular transport of internalized extracellular substances by PFP-driven cellular import and export in vesicular forms through endolysosomal pathways. This property is particularly important in vivo, where βγ-CAT could transport external substances such as water and Na+ to internal environments of toad tissues without disruption of the cellular tight junctions that maintain epithelial barrier functions. Interestingly, AQPs were identified in exosomes released in the presence of βγ-CAT (Fig. 5h, i), suggesting that βγ-CAT modulates water homeostasis by driving extracellular recycling and/or intercellular communication of these water channels. Alternatively, βγ-CAT-mediated exocytosis might function as a means for expulsion of noxious and indigestible solutes contained in fluid macropinocytosed by cells for water acquisition, like the expelling of dextran from toad UB epithelial cells (Fig. 5f, g). The role of βγ-CAT in expelling noxious substances engulfed by macropinocytosis via exocytosis induction and the cellular sorting mechanism involved are important future challenges.

Our study does not rule out the potential involvement of other physiological water transport mechanisms, such as AQPs and ion channels. In fact, our findings support the possibility that βγ-CAT may play an important role in toad B. maxima by participating in the regulation of classical water transport mechanisms, especially in the face of osmotic stress. It is generally understood that classic membrane integrated proteins such as AQPs and ion channels play fundamental roles in water transport and homeostasis[28,46]. Furthermore, the tight junction protein claudin-2 mediates the permeability to transepithelial water movements under osmotic or Na+ gradients through the paracellular pathway[47,48]. The present study has introduced a new player, a SELC protein βγ-CAT, into the network of water transport and homeostasis via the modulation of cellular material import and export through endolysosomal pathways. A cellular working model of the PFP βγ-CAT in promoting water uptake and maintaining was proposed in toad B. maxima (Fig. 6). In this model, water and Na+ can be imported by macropinocytosis induced by the PFP βγ-CAT. Meanwhile, the transmembrane channel formed by βγ-CAT can mediate the Na+ flow (Fig. 2b, c). Previous studies have shown that the channel formed by βγ-CAT can cause cation ion flux[39]. The PFP complex targets viral envelope to form pores that induced potassium and calcium ion efflux[26]. Unexpectedly, the average Na+ concentrations of βγ-CAT-containing exosomes were substantially lower than those of the control (Fig. 5j, k). This phenomenon suggested that the presence of channels formed by βγ-CAT can lead to rapid efflux of Na+ out of endolysosomes and/or exosomes. Thus, Na+ can be release into cytosol through βγ-CAT channels in endocytic organelles, which drives water release to the cytosol via AQPs (Fig. 4). This is similar to the situation of two-pore channels working in mammalian macrophages[42,43]. Enhanced exosome release mediated by βγ-CAT may promote the export of ions such as Na+ (Fig. 5l), and/or water, into the intercellular spaces below tight junctions, favoring water absorption into deeper cell environments. Meanwhile, membrane-integrated ion channels and AQPs in the basal plasma membrane of epithelial cells could

mediate the efflux of Na+ and water into internal tissues underneath these cells achieving the absorption/reabsorption of water into toad internal tissues[5,49]. Drinking water by macropinocytosis is energetically expensive[12]. However, the function of secretory PFP βγ-CAT in water acquisition and maintaining is necessary in light of various osmotic conditions that toad B. maxima has to face throughout its life cycle.

Secretory PFPs have been identified in organisms from all kingdoms of life[14–16], and af-PFPs are widely distributed in plants and animals[15–17]. Based on a wealth of experimental evidence on the af-PFP and TFF complex βγ-CAT from toad B. maxima[18–26], we have proposed the hypothesis of the SELC pathway mediated by a PFP, which drives cellular material import and export through endolysosomal pathways[17]. The present study provides further functional evidence to support our previous hypothesis. Regulated relying on environmental cues[17,18], SELC protein βγ-CAT has been shown to promote cellular intake and transport of environmental materials, like antigens, water with ions and plasma membrane components (see reference 24 and present study) relying on distinct cell contexts and surrounding, representing a hitherto unknown cell vesicular delivering system. These capacities raise the possibility that besides working in antigen presentation and water maintaining, βγ-CAT and/or its homologues could play active and fundamental roles in cell nutrient acquisition and metabolism flexibility, as previously proposed[17], which are intriguing subjects for further investigation. These PFPs and classic membrane integrated proteins (like solute carrier family members) can form two coordinated cell arms in extracellular nutrient sensing, sampling and acquiring. Obviously, these PFPs acting as SELCs should be essential to cells in those cases that classic membrane solute carriers are blockaded or even absent. Our findings will serve as clues for uncovering novel cellular strategies and the physiological roles of PFPs on material import and export via endolysosomal systems among cells and environments in living organisms, especially those in mammals.

In conclusion, the current work elucidated an unexpected role of a secretory PFP in water transport and homeostasis. βγ-CAT, an af-PFP and a TFF complex, induced and participated in macropinocytosis and exosome release in epithelial cells from B. maxima osmoregulatory organs. These effects could explain the role of the PFP complex in promoting the internalization, transport and release of water, Na+ and AQPs, working together in the maintenance of toad cell volume regulation and water homeostasis (Fig. 6). Together with membrane-integrated AQPs and ion channels, the action of secretory βγ-CAT can promote the overall achievement of water maintaining and balance, especially when the toad encounters hypertonic osmotic stress.

## Methods

**Animals.** Toads (B. maxima) were captured in the wild and raised at room temperature by feeding with live Tenebrio molitor. Toads with an average weight of 19 ± 5 g were used in experiments after fasting in isotonic Ringer's solution for 3 days. All the procedures and the care and handling of the animals were approved by the Institutional Animal Care and Use Committee at Kunming Institute of Zoology, Chinese Academy of Sciences (Approval ID: IACUC-OE-2021-05-001).

**Animal experiment.** Formulation of isotonic Ringer's solution (111.2 mM NaCl, 1.9 mM KCl, 1.1 mM CaCl$_2$, 2.4 mM NaHCO$_3$, 1.6 mM MgCl$_2$) was modified based on a previous report[50]. After fasting, toads in the hypertonic group were placed in hypertonic Ringer's solution (Ringer's solution with 222.4 mM NaCl) for 3 hours and their weight changes were recorded. Toads in the hypertonic/isotonic group were then transferred from hypertonic Ringer's solution to isotonic Ringer's solution for 3 hours and their weight changes were recorded. Weight changes of toads in isotonic Ringer's solution were recorded for 6 hours as a reference, termed the isotonic group. The skin of some toads was electrically stimulated (4.5 V DC, pulse duration 10 ms) for 3 minutes to deplete skin secretions, and toads were placed in isotonic Ringer's solution for 30 minutes. Their survival rates were then recorded after 48 hours in isotonic and hypertonic solutions.

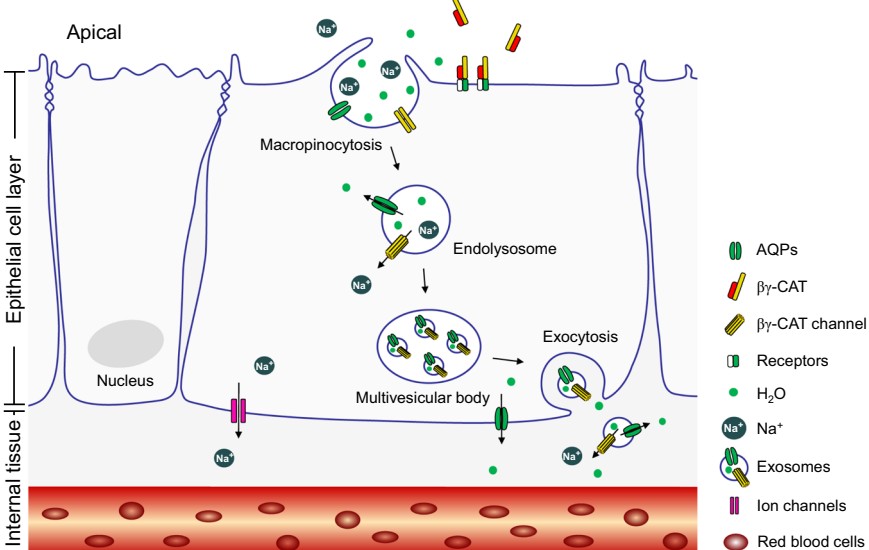

**Fig. 6 Proposed action model of βγ-CAT in water acquisition and maintaining.** βγ-CAT achieved intracellular vesicle transport and transcellular transport of water, Na$^+$ and AQP2 by promoting macropinocytosis (import) and exosome release (export). For a detailed description, see the test. Presentation of the epithelial cell layer and internal tissue is simplified. All clipart components are sourced from PowerPoint 2019 and Adobe illustrator CC 2019.

**Cell culture**. Toad epithelial cells were obtained by tissue digestion. Specifically, the UB, skin, and kidney tissues of *B. maxima* were dissected from five toads whose spinal cord had been destroyed. Tissues were further rinsed and stripped in Ringer's solution to remove residual blood, mucus and other impurities. They were then cut into pieces and washed twice in Ringer's solution, followed by oscillatory digestion with trypsin at room temperature for 40 minutes. The cells were separated with a 200-mesh sieve into a 50 mL centrifuge tube (Jet Biofil, Cat CFT011500). In addition, toad peritoneal cells were extracted from the peritoneal fluid of *B. maxima*. All cells were centrifugally enriched with 2,000 rpm for 5 min at 4 °C. The digestive cells in toad skin and UB tissues contain around 90% and 60% of epidermal cells, respectively, as analyzed by flow cytometry using an anti-pan cytokeratin AE1/AE3 monoclonal antibody (Invitrogen, Cat 53-9003-80).

Madin-Darby canine kidney (MDCK), Caco-2, HEK293 and T24 cell lines were purchased from Kunming Cell Bank, Chinese Academy of Sciences. The cells were cultured in DMEM/F-12 (Biological Industries, Cat 01-172-1 A) containing 10% fetal bovine serum (Biological Industries, Cat 04-001-1 A) and 1% levofloxacin hydrochloride and sodium chloride injection. All cell lines were cultured and grown to confluence in rat-tail collagen type I coated tissue culture flasks (Jet Biofil, Cat TCD010100) at 37 °C and 5% CO$_2$ in humid atmosphere.

**Purification of βγ-CAT**. βγ-CAT purification and purity analysis were carried out according to the previous description[19,22].

**Cell viability assay**. The MTS assay was used to detect the cytotoxicity of βγ-CAT. In brief, MDCK and T24 cell lines were seeded in 96-well plates at 1×10$^4$ cells per well and cultured overnight at 37 °C in 5% CO$_2$. Cells were incubated with the MTS reagent (Promega, Cat G3580) in the dark for 1 hour after treatment with βγ-CAT at room temperature for 2 hours. The absorbance of the toad cells culture supernatant was measured with 2×10$^6$ cells per well. The absorption was detected at 490 nm with the Infinite 200 Pro microplate reader (Tecan, Männedorf, Switzerland).

**Hemolytic activity**. 2×10$^7$ toad skin, UB, kidney or peritoneal cells were centrifugally enriched and then washed three times with 5 mL Ringer's solution until the supernatant had no hemolytic activity on human erythrocytes. The toad cell suspension was resuspended into four groups. The uncultured control group was centrifuged and the supernatant was collected. The other three groups were incubated at room temperature for 1.5 hours and centrifuged to collect the supernatant. One of them served as the cultured group. The remaining two batches were mixed with 50 μg mL$^{-1}$ rabbit antibodies or rabbit-derived anti-βγ-CAT antibodies, respectively, and incubated at room temperature for 30 minutes, while the uncultured and cultured groups were treated similarly using the same volume of Ringer's solution. Human erythrocytes were added to the supernatant at a concentration of 3%, incubated for 30 minutes at 37 °C, and then centrifuged at 2,000 rpm for 5 minutes. The absorption was detected at 540 nm with the Infinite 200 Pro microplate reader (Tecan, Männedorf, Switzerland).

**Cell diameter measurement**. Previously described methods were modified as appropriate[51]. Briefly, digested MDCK, Caco-2 and T24 cells were cultured in serum-free medium for 30 minutes at a concentration of 2×10$^6$ cells per mL before detection. To determine cell diameters, we treated MDCK, Caco-2 and T24 cells with hypertonic PBS (adding NaCl to 0.01 M PBS increased osmotic pressure to 400 mOsm) containing purified βγ-CAT at doses of 10 nM, 10 nM and 5 nM, respectively. Toad UB epithelial cells were incubated with or without 0.3 mM HgCl$_2$ for 10 minutes at a concentration of 2×10$^6$ cells per mL after incubating with 50 μg mL$^{-1}$ rabbit-derived anti-βγ-CAT antibodies for 30 minutes at room temperature. Diameter changes of toad UB epithelial cells were detected in isotonic or hypertonic Ringer's solution. The final concentration of all cells was 1×10$^6$ cells per mL. The average cell diameter was determined by a Countstar Automated Cell Counter (ALIT Life Science, Shanghai, China).

**Patch-clamp recordings**. Currents were recorded on HEK293 cells with the outside-out configuration of the patch-clamp technique at room temperature with a Multiclamp 700B amplifier and a Digidata 1550A analog-digital converter controlled by a pClamp10 software (Molecular Devices, San Jose, USA). The pipette solution contained 150 mM KCl, 10 mM HEPES, 1 mM EGTA, pH7.4; the bath solution contained 150 mM NaCl, 10 mM HEPES, pH7.4. The background (blank) and macroscopic current induced by 100 nM βγ-CAT were recorded under 500 millisecond ramp protocol between −100 mV to +100 mV every 2 second from a holding potential of 0 mV. For on replacement experiments, the pipette solution contained 150 mM NaCl, 10 mM HEPES, 1 mM EGTA, pH7.4; the bath solution contained 150 mM or 15 mM NaCl, 10 mM HEPES, pH7.4. All currents shown have been leak subtracted. Data were analyzed with pClamp10 and GraphPad Prism 8.

**Quantitative real-time PCR**. The mRNA levels of *βγ-CAT-α* and *βγ-CAT-β* in toad skin, UB and kidney were detected by qRT-PCR using a Hieff qPCR SYBR Green Master Mix (No Rox) kit (Yeasen, Cat 11201ES03). The cycle counts of target genes were normalized to those of *β-actin*. The mRNA levels of *FcRn* in toad cells from skin, UB, kidney and peritoneal cavity were also detected by RT-PCR. Primer sequences in this study are shown in Supplementary Table 1.

**RBITC-labeling of βγ-CAT**. The labeling reactions were performed in accordance with a previously described procedure with some modifications[52]. Purified βγ-CAT was dialyzed overnight in a cross-linking solution (NaHCO$_3$ 7.56 g, Na$_2$CO$_3$ 1.06 g, and NaCl 7.36 g dissolved in 1 L of water; pH 9.0) using a 3,500 Da dialysis membrane (Biosharp, Cat BS-QT-021), and 1 mg of RBITC (Sigma, Cat 283924) was dissolved in 1 mL of DMSO (Sigma, Cat D2650). The RBITC solution was slowly added to βγ-CAT to give βγ-CAT: RBITC = 1 mg:150 μg and the sample was incubated overnight at 4 °C, protected from light. NH$_4$Cl (50 mM) was added and the solution was incubated at 4 °C for 2 hours to terminate the reaction. The dialyzed samples were concentrated to 1 mg mL$^{-1}$ with PBS and stored at 4 °C in the dark.

**Immunofluorescence and HE staining**. Appropriate adjustments were used on a previous description[53]. Briefly, MDCK cells were grown to 90% on a 24-well glass slide. Paraformaldehyde-fixed sections of toad tissue samples and cell crawls were treated with 0.5% TritonX-100. After blocking for 2 hours in PBS containing 2% BSA at 37 °C, the tissues and cells were incubated with mouse-derived anti-AQP2 (Santa Cruz, Cat sc-515798) or rabbit-derived anti-βγ-CAT primary antibodies for 2 hours at 37 °C. The cells were incubated in the dark with fluorescence labeled secondary antibody for 1 hour at 37 °C after washing with PBS for three times. The samples were sealed with an anti-fluorescent quench agent containing DAPI. The localization of βγ-CAT or BmAQP2 in toad tissues was determined by using rabbit-derived anti-AQP2 (ImmunoWay, Cat YT0290) and mouse-derived anti-βγ-CAT primary antibodies.

MDCK cells were treated with a dose of 10 nM RBITC-βγ-CAT, in 0.01 M PBS with or without 1 mg mL$^{-1}$ FITC-label dextran (Sigma, Cat 46945) for 15 minutes at 37 °C. After washing with PBS for three times, the cells were fixed in 4% paraformaldehyde for 15 minutes, and then treated with 0.5% TritonX-100 for 15 minutes. The samples were sealed with an anti-fluorescent quench agent containing DAPI.

Images of Fig. 4b were acquired by a Nikon A1 confocal laser microscope system (Nikon, Tokyo, Japan). Images of Fig. 1g, h, Fig. 4a and supplement Fig. 3g were acquired by a Zeiss LSM 880 microscope system (Carl Zeiss, Oberkochen, Germany).

**Isolation of exosomes**. The method to isolate exosomes was optimized based on a previous description[24]. Briefly, MDCK and T24 cells were grown to 90% on 100-mm cell culture dishes. The cells were cultured in twenty cell culture dishes with and without 10 nM or 5 nM βγ-CAT for 3 hours at 37 °C. Cell supernatant (200 mL) was collected for gradient centrifugation (300 × g for 30 minutes; 500 × g for 30 minutes; 2,000 × g for 30 minutes; 10,000 × g for 40 minutes) at 4 °C to remove residual cells, debris and microvesicles. Exosomes were obtained by ultracentrifugation at 100,000 × g with rotor P100AT2 for 2 hours at 4 °C, and then resuspended in PBS, and enriched again in the CP100WX preparative ultra-centrifuge (HATACHI, Tokyo, Japan). The enriched exosomes were stored at −80 °C for follow-up experiments.

Healthy toads B. maxima were used to take tissue and grind it to obtain cells. 2×10$^7$ cells obtained from three toads were cultured in 10 mL Ringer's solution containing either 50 nM βγ-CAT, 50 μg mL$^{-1}$ rabbit IgG or 50 μg mL$^{-1}$ rabbit-derived anti-βγ-CAT antibodies for 3 hours at room temperature. Exosomes were extracted with Hieff Quick exosome isolation kit (Yeasen, Cat 41201-A). Cell supernatant was obtained by a series of gradient centrifugations (500 × g for 10 minutes; 3,000 × g for 10 minutes). It was thoroughly mixed with a quarter volume of the reagent at 4 °C for 2 hours and then centrifuged at 10,000 × g for 1 hour to collect exosomes. After resuspension in Ringer's solution, the liquid containing exosomes was centrifuged at 100,000 × g with rotor P100AT2 for 2 hours at 4 °C, and repeated once in a CP100WX preparative ultracentrifuge (HATACHI, Tokyo, Japan). The enriched exosomes were stored at −80 °C for follow-up experiments.

**Flow cytometry**. 2×10$^5$ MDCK and T24 cells were incubated with 100 μg mL$^{-1}$ 70 kDa FITC-label dextran (Sigma, Cat 46945) or Lucifer Yellow (Sigma, Cat L0144) in the dark at 37 °C for 30 minutes with and without 10 nM or 5 nM βγ-CAT, respectively. The samples were followed by fluorescence detection of FITC or AmCyan. In each sample, 1×10$^4$ single cells were analyzed. 2×10$^6$ toad UB epithelial cells and peritoneal cells were treated with 50 nM βγ-CAT, while 100 nM βγ-CAT was used for 2×10$^6$ toad skin and kidney cells. In the test using immunodepletion of endogenous βγ-CAT, toad cells were incubated with 50 μg mL$^{-1}$ rabbit-derived anti-βγ-CAT antibodies for 30 minutes before the above protocol was carried out. To test the non-specific effect of exogenous rabbit IgG, MDCK cells were incubated with 50 μg mL$^{-1}$ of rabbit IgG or rabbit-derived anti-βγ-CAT antibody for 30 minutes prior to the above protocol. During the inhibitor experiment, the cells were first incubated with 100 μM EIPA (MedChemExpress, Cat HY-101840A) or 20 μM wortmannin (Sigma, Cat 681675) for 1 hour at 37 °C. In addition, 2×10$^6$ digested toad UB epithelial cells were cultured in vitro for 3 hours and co-incubated with 500 ng mL$^{-1}$ propidium iodide (BD Pharmingen, Cat 556547) for 10 minutes at room temperature. The fluorescence was recorded using LSR Fortessa cell analyzer (Becton Dickinson, Franklin Lakes, NJ, USA). The data were analyzed by FlowJo 10 and GraphPad Prism 8.

**Flow cytometry for nanoparticle analysis**. Exosomes of the toad UB epithelial cells were isolated as described in "**Isolation of exosomes**". Exosomes or 2×10$^7$ toad UB epithelial cells were cultured in medium containing 1 mg mL$^{-1}$ FITC-label dextran with or without 50 nM βγ-CAT at room temperature for 3 hours. Exosomes of the toad UB epithelial cells were enriched and residual dextran was washed off using the method described in "**Isolation of exosomes**". All experiments were done in the dark. The fluorescence and particle size distribution were recorded using Flow NanoAnalyzer (NanoFCM, Xiamen, China). The data were analyzed by FlowJo 10 and GraphPad Prism 8.

**Transmission electron microscopy (TEM) and immunoelectron microscopy (IEM)**. The experiments were based on the previous report[54]. Tissue samples were fixed overnight with 0.1 M PB (pH 7.4) containing 3% paraformaldehyde 0.1% glutaraldehyde at 4 °C, washed with 0.1 M PB four times for 15 minutes, and then washed with 0.1 M glycine in 0.1 M PB for 30 minutes at 4 °C. After ethanol gradient dehydration, the sample was embedded in LR white resin (Sigma, Cat L9774) and polymerized at 55 °C for 24 hours. 100-nm ultrathin sections were prepared using an EM UC7 ultramicrotome (Leica Microsystems, Wetzlar, Germany) and loaded onto 200-mesh Ni grids (EMCN, Cat BZ10262Na). Enriched exosomes were directly attached to Ni grids for 10 minutes before experimental treatment. The samples were washed with deionized water for 2 minutes and then blocked with 1% BSA for 5 minutes. After overnight incubation with mouse-derived anti-βγ-CAT or rabbit-derived anti-AQP2 (ImmunoWay, Cat YT0290) primary antibodies at 4 °C, the samples were washed with deionized water 10 times. Then, the samples were incubated with 5-nm or 10-nm colloidal gold-conjugated secondary antibody (Sigma, Cat G7527 and G7402) at room temperature for 2 hours, and washed 10 times at room temperature. Exosomes were stained with 2% uranyl acetate for 3 minutes, and sections were stained with 2% uranyl acetate for 7 minutes and lead citrate for 5 minutes before observation by employing a JEM 1400 plus transmission electron microscope at 100 kV.

**Intracellular and exosome sodium detection**. 2×10$^7$ toad UB epithelial cells enriched by centrifugation were resuspended with Ringer's solution (N-Methyl-D-glucamine was used instead of NaCl) after digestion and washing 3 times. MDCK cells were cultured to 90% on 100-mm cell culture dishes, and three cell culture dishes in each group were used in this experiment. Toad UB epithelial cells and MDCK cells were cultured for 3 hours with and without 50 nM or 10 nM βγ-CAT, respectively. The cells were collected, and lysed with deionized water. Sodium concentrations in MDCK exosomes were measured in isotonic or hypertonic assays using forty or twenty cell culture dishes in each group, respectively. MDCK exosomes were isolated as described in "**Isolation of exosomes**". The samples were frozen in liquid nitrogen and then slowly melted at room temperature, and repeated twice. Ultrasonic cell crushing (power 200 W, work every 10 seconds for 5 seconds, 40 cycles) was used to release the sample until the liquid was transparent. Impurities in samples were removed by gradient centrifugation (2000 × g for 30 minutes; 10,000 × g for 1 hour). After the sample was filtered with a 0.22 μm filter, the concentration of sodium ions in the samples was determined by iCAP6300 inductively coupled plasma optical emission spectrometer (Thermo Fisher Scientific, Waltham, United States of America).

**Nanoparticle tracking analysis (NTA)**. All exosome samples were diluted to around 1×10$^7$ particles per mL with PBS. The particle size and concentration of exosomes were measured by ZetaView PMX 110 (Particle Metrix, Meerbusch, Germany), and the data were analyzed using the software ZetaView 8.04.02.

**Western blotting**. To measure the protein level of βγ-CAT or AQP2, toad UB epithelial cells, MDCK, Caco-2, and T24 cells were treated for 15 minutes with or without 50 nM, 10 nM, 10 nM and 5 nM βγ-CAT, respectively. Total Akt and phospho-Akt were detected by treating MDCK cells with or without 10 nM βγ-CAT for 30 minutes or 60 minutes. Exosomes from toad UB epithelial cells were washed with Ringer's solution and harvested by RIPA lysis buffer on ice. The rabbit-derived anti-βγ-CAT, rabbit-derived anti-AQP2 (ImmunoWay, Cat YT0290), mouse-derived anti-CD63 (Abcam, Cat ab193349), rabbit-derived anti-TSG101 (Proteintech, Cat 28283-1-AP), rabbit-derived anti-flotillin 1 (Proteintech, Cat 11571-1-AP), rabbit-derived anti-PI3 Kinase p85 (CST, Cat 4257), rabbit-derived anti-phospho-PI3 Kinase p85 (CST, Cat 4228), rabbit-derived anti-Rac1/2/3 (CST, Cat 2465), rabbit-derived anti-total Akt (CST, Cat 4691) and rabbit-derived anti-phospho-Akt-S473 (CST, Cat 4060) primary antibodies were used in these experiments.

**Sequence alignment and sequence analysis**. The evolutionary history was inferred by using the Maximum Likelihood method based on the Poisson correction model[55]. Initial tree(s) for the heuristic search were obtained automatically by applying Neighbor-Join and BioNJ algorithms to a matrix of pairwise distances estimated using a JTT model, and then selecting the topology with superior log likelihood value[56]. Evolutionary analysis was conducted in MEGA7. In addition, sequence alignment analysis of AQPs in toad B. maxima and other species was done by Clustal Omega.

**Statistical analysis**. Animal survival data were analyzed by the Gehan-Breslow-Wilcoxon test. All other data were analyzed using the standard unpaired t-test. Differences with P values < 0.05 were considered statistically significance. All statistical analyses were conducted using GraphPad Prism 8 software.

**Reporting summary**. Further information on research design is available in the Nature Research Reporting Summary linked to this article.

## Data availability

The sources for all data supporting the results of this study are cited in the main text, in Supplementary figures, and in Supplementary table 1. Uncropped and unedited blot/gel images as Supplementary fig. 5 at the Supplementary information. The datasets generated during and/or analysed during the current study are available in the Dryad Data repository (https://doi.org/10.5061/dryad.0p2ngf226)[57]. All clipart components in Fig. 6 are sourced from PowerPoint 2019 and Adobe illustrator CC 2019.

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

## Acknowledgements

This work was supported by grants from the National Natural Science Foundation of China (grant numbers 31872226, U1602225 and 31572268) and the Yunling Scholar Program to Yun Zhang. We would like to thank the Kunming Biological Diversity Regional Center of Instrument, Kunming Institute of Zoology, Chinese Academy of Sciences for our electron microscopy and we would be grateful to Ying-Qi Guo for her help of making EM samples.

## Author contributions

These authors contributed equally to this work: Z.Z., and Z.H.S. Correspondence to Y.Z. Y.Z., Z.Z., and Z.H.S conceived of and conceptualized the study; Z.Z., Z.H.S., and C.J.Y did the experiments, Z.Z., Y.Z., Z.H.S, and C.J.Y analyzed and interpreted the data; Z.Z., Y.Z., Z.H.S wrote the manuscript; Y.Z., Z.Z., Z.H.S and C.J.Y critically revised the manuscript for important intellectual content.

## Competing interests

The authors declare no competing interests.
