## [Peer Review File · Communications Biology]

Reviewers' comments:

Reviewer #1 (Remarks to the Author):

Zhao et al. report a novel function for the pore-forming bg-CAT complex in toad water maintenance. The authors show that expression and localization of bg-CAT are regulated by osmotic stress in vivo and counteract cell volume changes in toad and mammalian cells in culture. The authors further provide evidence that bg-CAT promotes macropinocytic uptake of extracellular fluid and solutes, and release of aquaporin and Na⁺-containing exosomes, which could explain its function in regulating water maintenance. This is an interesting paper that investigates the poorly understood property of water-permissive amphibian epithelia to adjust to environments of different osmolarities. The results are of broad interest to readers interested in animal physiology and – by providing insights into novel physiological functions of macropinocytosis and exosome formation – for cell biologist who work on these processes in any other contexts. The manuscript addresses this process at multiple steps and from multiple angles in an overall convincing way. Some of the conclusions would benefit from further experimental support, as detailed below.

Major points:

1. Fig. 3b, c: Treatment of cells with rabbit IgG seems to cause a substantial increase in dextran and LY uptake. Do the authors have an explanation why treatment with an unspecific antibody would increase macropinocytosis? Moreover, anti-bg-CAT-treated cells display less macropinocytosis than control IgG-treated cells, but more than vehicle-treated cells. Based on this the authors conclude that immunodepletion of bg-CAT blocks macropinocytosis. Can they rule out that this antibody simply induces less macropinocytosis than the control IgG?
2. Fig. S4B: Treating cells with control IgG causes a substantial increase in exosome release, which confounds interpretation of the results (see above).
3. Fig. S3: It is difficult for me to tell what exactly the images for dextran uptake show. In MDCK cells (S3G), dextran mostly colocalizes with the nuclear dye, which contradicts its presence in macropinosomes. Maybe this is due to z-projection? It would be helpful if the authors showed higher magnification images to better visualize macropinosomes. Images for macropinocytosis inhibitor-treated cells would also be helpful.
4. Fig. 4: The AQP staining in vivo looks very convincing, but I am surprised by the subcellular localization in cultured cells. Is it expected to see AQP in a few large patches per cell?
5. The authors propose that bg-CAT-induced macropinocytosis and exosome release regulate water maintenance. The manuscript presents circumstantial evidence for this model, and it would be interesting to know whether blocking macropinocytosis or exosome directly affects cell size depending on osmotic environment. Due to the overall novelty of the paper, in my opinion it is not required to show this in detail, but the authors should be careful to separate the discussion of results from speculation.

Minor points:

1. It would be very interesting to know whether medium osmolarity influences macropinocytosis.
2. 70 kDa dextran is a bona fide marker for macropinosomes, but LY is a marker for fluid-phase endocytosis (including macropinocytosis).
3. Fig. 1g,h: Please include high magnification images for the immunostainings.
4. Fig. 2a: Please provide loading controls.
5. Fig. 3i: The change in phospho-Akt is very subtle. It would be good to provide further evidence that bg-CAT increases PI3-kinase signaling, e.g. other concentrations, more antibodies, cell lines etc. Also, in the graph please set the origin to 0 to better allow the reader to assess the effect sizes.

6. Fig. 5c: Flotillion-1 – I guess this is flotillin-1?

7. Fig. S3 E-G: The brightfield images are difficult to see – please improve contrast.

Reviewer #2 (Remarks to the Author):

Brief summary of the manuscript

This is a study investigating possible osmoregulatory functions of a so called 'secretory membrane complex', $\beta\gamma$ -CAT, which is formed by interaction of BmALP1 with BmTFF3 to form an active pore forming protein (PFP). Previous studies led this team of investigators to conclude that BmTFF3 acts as an extracellular chaperon that stabilizes the BmALP1 monomer exporting the pore forming protein to its proper membrane position. Here, they propose a novel role for $\beta\gamma$ -CAT in toad water metabolism. This conclusion is based upon a combination of experimental observations and speculations.

(i) $\beta\gamma$ -CAT was found to be constitutively expressed in toad osmoregulatory organs and inducible under imposed variation of environmental osmotic conditions.

(ii) 'Deletion of toad skin secretions', in which $\beta\gamma$ -CAT is a major component, increased animal mortality under hypertonic stress.

(iii) The protein induced and participated in macro-pinocytosis in vivo and in vitro.

(iv) In response to extracellular hyperosmotic treatment, $\beta\gamma$ -CAT stimulated macro-pinocytosis assumed to facilitate water intake and enhanced exosomes release, simultaneously with regulating aquaporins distribution.

(v) These findings/interpretations lead to the authors' conclusion that besides the well-known aquaporins, a secretory pore-forming protein facilitates toad water balance via a postulated 'macro-pinocytosis induction and exocytosis modulation' in the skin epithelium of animals submitted to osmotic stress.

Overall impression of the work

The authors successfully master a number of molecular and cell biological methods, which are applied to elucidate regulation of the water metabolism of the terrestrial toad *Bufo maxima*. The study constitutes an imbalance of experimental approaches at the cell/molecular level on the one hand and the physiology of the intact animal on the other; the latter being represented, by and large, upon 'survival studies' and 'whole body weight measurements'. The 'skin' of the toad is considered a single-layered epithelium disregarding its multi-component anatomy of individual osmoregulatory significance.

Specific comments, with recommendations for addressing each comment

1. Page 5, line 4-5. The two references mentioned are highly relevant. However recently, based upon considerable experimental evidence it has been concluded that amphibian skin plays a dual role in amphibian osmoregulation, see

Larsen EH (2021) Dual skin functions in amphibian osmoregulation. *Comparative Biochemistry and Physiology A. Molecular and Integrative Physiology* 253: 110869

It is the thesis of this review (and of previous recent studies by this author's laboratory) that water balance of amphibians on land depends on integrated functions of the heterocellular surface epithelium, the subepidermal exocrine glands, and the cutaneous surface liquid.

I urge the authors to consider the above conceptual framework when discussing the function of skin in amphibian osmoregulation.

2. The authors' multi-experimental approaches are mostly based upon cellular and biochemical methods, which are handled professionally. This has resulted in a number of solid observations of interest for their own sake. However, it is also clear to me that taken together several individual results do not provide interrelated interpretations of consistent significance for the understanding of the macroscopic osmotic function of the skin. Rather than accepting this openly, the authors tend to speculate on their physiological importance. Serving as typical example, on page 15 line 450-457 the authors write:

"Obviously, these SELC proteins should be essential to cells in those cases that classic membrane solute carriers are blockaded or even absent. Our findings will serve as clues for uncovering the physiological roles of PFPs in other living organisms. Specifically, proliferating embryonic cells of

oviparous animals depend entirely on catabolism of extracellular macromolecules deposited in the egg white and yolk. If macropinocytosis-driven by a PFP plays a role in oviparous animal embryonic development is an interesting subject for further investigation.”

Rather than merely asking the authors to clean their manuscript from this type of speculations remote from their actual experimental observations, I suggest that they make an effort to reconsider their general experimental protocol, which is a comparison between effects of hyperosmotic and isoosmotic exposures of the skin, respectively. More specifically, I suggest that the authors take into account that on land the skin is exposed to slightly hypoosmotic environment in terms of the cutaneous surface liquid. Exposure of the toad to external isoosmotic conditions might well simulate the cutaneous surface liquid, so that the experimental results obtained thus obtained can be as is. However, the other physiological challenge of the skin is experienced when the toad is in freshwater. This is a strongly hypoosmotic, not an hyperosmotic exposure. In other words, I ask for new experiments allowing for comparisons between these environmental conditions. I am aware that one of the incentives for this study is the observation that the ill-defined 'deletion of toad skin secretions', in which $\beta\gamma$ -CAT is a major component, increased animal mortality under hypertonic stress. But taking the fact that few amphibian species (e.g., the green toad *B. viridis*) tolerate exposure to hypertonic saline, I find the above observation to be a curiosity rather than a response of importance for a general description of the toad's survival strategy.

Peer Review File

Manuscript Title: A pore-forming protein drives macropinocytosis to facilitate toad water maintaining

Your manuscript entitled "A pore-forming protein drives macropinocytosis to facilitate toad water maintaining" has now been seen by 2 referees, whose comments are appended below. You will see from their comments copied below that while they find your work of potential interest, they have raised quite substantial concerns that must be addressed. In light of these comments, we cannot accept the manuscript for publication, but would be interested in considering a revised version that addresses these serious concerns.

We hope you will find the referees' comments useful as you decide how to proceed. Should further experimental data or analysis allow you to address these criticisms, we would be happy to look at a substantially revised manuscript. However, please bear in mind that we will be reluctant to approach the referees again in the absence of major revisions.

In particular, we would like to raise attention to some key comments from the referees. Reviewer 1 believes that some additional experimental work is necessary to solidify the conclusions, and we agree with this. Reviewer 2 also feels this way, but also suggests discussing the results in light of a certain framework. We are willing to be flexible on this point, as the overall framework is not the sole approach present in the field, but a detailed response to the reviewer about these points should be provided if changes are not incorporated.

Response:

Thank you for your valuable advice. We greatly appreciate your consideration of our manuscript. We have revised our original manuscript according to the suggestions of the reviewers. We hope this version of the manuscript will give you satisfaction.

Reviewer Comments & Author Response

The following is a point-by-point response to the reviewers' comments:

Reviewers' comments:

Reviewer #1 (Remarks to the Author):

Zhao et al. report a novel function for the pore-forming bg-CAT complex in toad water maintenance. The authors show that expression and localization of bg-CAT are regulated by osmotic stress in vivo and counteract cell volume changes in toad and mammalian cells

in culture. The authors further provide evidence that bg-CAT promotes macropinocytic uptake of extracellular fluid and solutes, and release of aquaporin and Na⁺-containing exosomes, which could explain its function in regulating water maintenance. This is an interesting paper that investigates the poorly understood property of water-permissive amphibian epithelia to adjust to environments of different osmolarities. The results are of broad interest to readers interested in animal physiology and – by providing insights into novel physiological functions of macropinocytosis and exosome formation – for cell biologists who work on these processes in any other contexts. The manuscript addresses this process at multiple steps and from multiple angles in an overall convincing way. Some of the conclusions would benefit from further experimental support, as detailed below.

Response:

Thanks for your valuable comments. We have revised our manuscript carefully according to your suggestion. The detailed changes are listed below.

Major points

General comments:

1. Fig. 3b, c: Treatment of cells with rabbit IgG seems to cause a substantial increase in dextran and LY uptake. Do the authors have an explanation why treatment with an unspecific antibody would increase macropinocytosis? Moreover, anti-bg-CAT-treated cells display less macropinocytosis than control IgG-treated cells, but more than vehicle-treated cells. Based on this the authors conclude that immunodepletion of bg-CAT blocks macropinocytosis. Can they rule out that this antibody simply induces less macropinocytosis than the control IgG?

Response:

We thank the referee for pointing out these issues. For the above issues, we have described them in detail in the discussion of the latest manuscript. Please see line 397, page 15. Here we provide a brief summary:

1. The rabbit IgG (control antibody) and anti-βγ-CAT antibody were derived from rabbits. The method of obtaining the rabbit IgG was to isolate and purify the IgG of rabbits before self-immunization, while anti-βγ-CAT antibody was to isolate and purify the IgG after immunizing rabbits with purified βγ-CAT. In the terms of structure, antigen-binding fragments (Fab) of the two types of IgG are different. Functionally, the two IgG antibodies differ in their ability to bind βγ-CAT and block its biological function, so we think this is a strictly controlled trial. Aiming at the above problems, we supplemented experiments to support our conclusions. Since non-specific rabbit IgG increased pinocytosis in toad cells, it was necessary to evaluate the effect of rabbit IgG and anti-βγ-CAT antibodies on

pinocytosis. We found that both two non-specific antibodies increased the pinocytosis capacity of MDCK cells to the same extent when cells were treated with two antibodies at the same concentration (Fig. A in this file). In addition, rabbit IgG and $\beta\gamma$ -CAT co-stimulated MDCK cells to increase pinocytosis (Fig. B in this file). These suggested that non-specific rabbit IgG can promote pinocytosis with active $\beta\gamma$ -CAT. Due to the co-stimulation of exocytosis by non-specific IgG and endogenous $\beta\gamma$ -CAT, the number of exosomes in the rabbit IgG group was significantly higher than that in the other two groups (Fig. 5d and Supplementary Fig. 4b). We validated this combined stimulative effect on MDCK cells with rabbit IgG and anti- $\beta\gamma$ -CAT antibodies. Both non-specific antibodies can equally increase the release of MDCK-derived exosomes, but this phenomenon was more significant in rabbit IgG and exogenous $\beta\gamma$ -CAT co-stimulation (Fig. C in this file).

2. In addition to antigen-antibody recognition, there are receptors for IgG in most cells such as Fc γ R family and FcRn. Both types of receptors bind to the crystalline fragments (Fc) of IgG and stimulate endocytosis and exocytosis^{1,2}. For example, pinocytosis and exocytosis, which depend on the combination of IgG and FcRn, can achieve both the recycling of IgG and the delivery of substances in cells. Therefore, this effect of IgG can lead to increased intake of dextran and LY. In fact, we found the presence of the FcRn-like gene in cells of various osmotic regulatory organs of toad *Bombina maxima* by PCR (Fig. D in this file). Because nonspecific IgG cannot specifically bind to $\beta\gamma$ -CAT and cause $\beta\gamma$ -CAT inactivation, there are actually two pinocytic stimulants in the rabbit IgG group, rabbit IgG and endogenous active $\beta\gamma$ -CAT, which may result in significantly higher pinocytosis levels than the other groups (Fig. 3b, c and Supplementary Fig. 3b). Due to the blocking of endogenous $\beta\gamma$ -CAT by the specific Fab of anti- $\beta\gamma$ -CAT antibody, the decreased pinocytosis ability of the cells in the anti- $\beta\gamma$ -CAT antibody group reflects that $\beta\gamma$ -CAT can promote pinocytosis. However, rabbit-derived anti- $\beta\gamma$ -CAT IgG antibody itself also interacts with Fc receptors and stimulates pinocytosis, so it is possible that the pinocytosis capacity of the anti- $\beta\gamma$ -CAT antibody group is higher than that of the vehicle group. In fact, we believe that the strict control trail is formed between the rabbit IgG group and anti- $\beta\gamma$ -CAT antibody group in figure. Since there are multiple variables between the anti- $\beta\gamma$ -CAT antibody group and the vehicle group, $\beta\gamma$ -CAT activity and IgG activity, we do not have a comparison between the two groups. However, the existence of vehicle group is the basis of the rationality of the experiment, which reflects the pinocytosis ability and cell vitality.

3. The ability of $\beta\gamma$ -CAT to promote macropinocytosis is based on the results of a variety of experiments, including but not limited to antibody tests. For example, we also proved the ability of exogenously adding purified $\beta\gamma$ -CAT to enhance macropinocytosis in mammalian cells and toad cells. And we've shown that in inhibitor trials. We think the combined results of these experiments reflect the true function of $\beta\gamma$ -CAT.

General comments:

2. Fig. S4B: Treating cells with control IgG causes a substantial increase in exosome release, which confounds interpretation of the results (see above).

Response:

Thanks for your suggestion. For the questions, we have described them in detail in the discussion of the latest manuscript. Please see line 437, page 17.

As mentioned above in the response, it has been widely demonstrated that the binding of IgG to membrane receptors can stimulate cellular endocytosis and exocytosis, which was also observed in our experiments. We have a few more points to emphasize here. Due to the presence of IgG and endogenous active $\beta\gamma$ -CAT in the rabbit IgG group, the overall exocytosis stimulation ability was significantly stronger than other groups. Compared with the rabbit IgG group, the decrease of exosome number reflected the loss of $\beta\gamma$ -CAT function in the anti- $\beta\gamma$ -CAT antibody group. In addition, we have demonstrated that exogenously adding purified $\beta\gamma$ -CAT can promote exosome release in mammalian cells and toad cells. Similarly, we do not consider the control and anti- $\beta\gamma$ -CAT groups to be directly comparable because of multiple variable differences in IgG and $\beta\gamma$ -CAT activity.

General comments:

3. Fig. S3: It is difficult for me to tell what exactly the images for dextran uptake show. In MDCK cells (S3G), dextran mostly colocalizes with the nuclear dye, which contradicts its presence in macropinosomes. Maybe this is due to z-projection? It would be helpful if the authors showed higher magnification images to better visualize macropinosomes. Images for macropinocytosis inhibitor-treated cells would also be helpful.

Response:

Thanks for your valuable suggestion. For figure S3, we have refined this experiment and used a higher resolution two-photon microscope for image acquisition. We still use 70 kDa FITC-dextran to indicate macropinosomes, which is considered reasonable. It was found that dextran was scattered in the cells. After $\beta\gamma$ -CAT treatment, intracellular dextran was increased and co-located with $\beta\gamma$ -CAT. At the same time, $\beta\gamma$ -CAT may affect the distribution of dextran-indicated macropinosomes in cells. However, it is difficult to achieve long-term culture of primary cells from toad *Bombina maxima* in vitro, although we have tried many methods. Thus, nucleus of toad cells appear larger in cells with no extended morphology, which may lead to the inevitable overlap between dextran and nucleus. In addition, we have analyzed the effect of $\beta\gamma$ -CAT on dextran uptake in cells using macropinocytosis inhibitors and found that inhibitors inhibit the $\beta\gamma$ -CAT-induced internalization of dextran (Fig. 3g, h). Therefore, the inhibitors were no longer used for fluorescence imaging. Please see figure S3 in the supplemental information, page 14.

General comments:

4. Fig. 4: The AQP staining in vivo looks very convincing, but I am surprised by the subcellular localization in cultured cells. Is it expected to see AQP in a few large patches per cell?

Response:

Thanks for your careful review and valuable suggestion. After observing the relationship between AQP2 and $\beta\gamma$ -CAT in toads, we wanted to further verify the localization relationship between AQP2 and $\beta\gamma$ -CAT in mammalian cells. We have refined this experiment and used a higher resolution two-photon microscope for imaging. Our latest results also support the co-location of AQP2 and $\beta\gamma$ -CAT in MDCK. We have added this data to the latest manuscript and removed AQP3 data to make the structure of the manuscript more reasonable. Please see figure 4, page 35.

General comments:

5. The authors propose that $\beta\gamma$ -CAT-induced macropinocytosis and exosome release regulate water maintenance. The manuscript presents circumstantial evidence for this model, and it would be interesting to know whether blocking macropinocytosis or exosome directly affects cell size depending on osmotic environment. Due to the overall novelty of the paper, in my opinion it is not required to show this in detail, but the authors should be careful to separate the discussion of results from speculation.

Response:

Thanks for your valuable suggestion. At present, the widely accepted theory of cell size and osmotic regulation is based on the participation of aquaporins (AQPs) and ions (like Na^+ 、 K^+)³⁻⁵. And we strongly agree with this view. In fact, we also highlighted the effect of $\beta\gamma$ -CAT on AQPs and ions in Figure 2, Figure 4 and Figure 5. We believe that the vesicle-dependent transport of water and ions by $\beta\gamma$ -CAT is an economical way to facilitate toad water maintaining in the face of osmotic challenges. In particular, the urinary bladder and kidney are confronted with hypertonic fluid environment at any time under physiological conditions. In addition, we have adjusted and revised the discussion of the manuscript to make it more reasonable. Please see line 369, page 14.

Minor points:

General comments:

1. It would be very interesting to know whether medium osmolarity influences macropinocytosis.

Response:

Thanks for your meaningful and careful review. This is really a question worth thinking about. Based on our results, we have reason to believe that osmotic pressure may be related to pinocytosis. However, it must be pointed out that the relationship between AQPs, ion channels and pinocytosis is difficult to measure and explain whether pinocytosis is a passive or active biological process when the cells face osmotic challenges.

General comments:

2. 70 kDa dextran is a bona fide marker for macropinosomes, but LY is a marker for fluid-phase endocytosis (including macropinocytosis).

Response:

Thanks for your careful review and valuable suggestion. We have revised it in the latest manuscript. Please see line 240, page 10.

General comments:

3. Fig. 1g,h: Please include high magnification images for the immunostainings.

Response:

Thanks for your careful review. We have revised it in the latest manuscript. Please see figure 1g and h, page 29.

General comments:

4. Fig. 2a: Please provide loading controls.

Response:

Thanks for your valuable suggestion. We have added loading controls of western blotting to the latest edition of the manuscript. Please see figure 2a, page 31.

General comments:

5. Fig. 3i: The change in phospho-Akt is very subtle. It would be good to provide further evidence that bg-CAT increases PI3-kinase signaling, e.g. other concentrations, more antibodies, cell lines etc. Also, in the graph please set the origin to 0 to better allow the reader to assess the effect sizes.

Response:

Thanks for your careful review. We added the influence of $\beta\gamma$ -CAT on PI3K, Rac123 and Akt in the latest version of the manuscript. Please see line 261, page 11; figure 3i, page 33.

General comments:

6. Fig. 5c: Flotillion-1 – I guess this is flotillin-1?

Response:

Thanks for your careful review and valuable suggestion. Because of our oversight, there really is a mistake here. We have revised it in the latest manuscript. Please see line 300, page 12; figure 5c, page 36.

General comments:

7. Fig. S3 E-G: The brightfield images are difficult to see – please improve contrast.

Response:

Thanks for your valuable suggestion. We have refined this experiment in the latest manuscript and adjusted the way the data is presented. Please see figure S3 e-g in the supplemental information, page 14.

Reviewer #2 (Remarks to the Author):

Brief summary of the manuscript

This is a study investigating possible osmoregulatory functions of a so called 'secretory membrane complex', $\beta\gamma$ -CAT, which is formed by interaction of BmALP1 with BmTFF3 to form an active pore forming protein (PFP). Previous studies led this team of investigators to conclude that BmTFF3 acts as an extracellular chaperon that stabilizes the BmALP1 monomer exporting the pore forming protein to its proper membrane position. Here, they propose a novel role for $\beta\gamma$ -CAT in toad water metabolism. This conclusion is based upon a combination of experimental observations and speculations.

(i) $\beta\gamma$ -CAT was found to be constitutively expressed in toad osmoregulatory organs and inducible under imposed variation of environmental osmotic conditions.

(ii) 'Deletion of toad skin secretions', in which $\beta\gamma$ -CAT is a major component, increased animal mortality under hypertonic stress.

(iii) The protein induced and participated in macro-pinocytosis in vivo and in vitro.

(iv) In response to extracellular hyperosmotic treatment, $\beta\gamma$ -CAT stimulated macro-pinocytosis assumed to facilitate water intake and enhanced exosomes release, simultaneously with regulating aquaporins distribution.

(v) These findings/interpretations lead to the authors' conclusion that besides the well-known aquaporins, a secretory pore-forming protein facilitates toad water balance via a postulated 'macro-pinocytosis induction and exocytosis modulation' in the skin epithelium of animals submitted to osmotic stress.

Overall impression of the work

The authors successfully master a number of molecular and cell biological methods, which are applied to elucidate regulation of the water metabolism of the terrestrial toad *Bufo maxima*. The study constitutes an imbalance of experimental approaches at the cell/molecular level on the one hand and the physiology of the intact animal on the other; the latter being represented, by and large, upon 'survival studies' and 'whole body weight measurements'. The 'skin' of the toad is considered a single-layered epithelium disregarding its multi-component anatomy of individual osmoregulatory significance.

Response:

Thank you very much for your positive and valuable comments on our research work. Due to the interference of our previous manuscript style and pattern diagram in figure 6, the understanding of the manuscript may be different from what we want to express. There are several points we would like to point out here.

1. Although $\beta\gamma$ -CAT was first identified in skin secretions of toad *Bombina maxima*, it is undeniable that $\beta\gamma$ -CAT is widely distributed in a variety of osmotic regulatory organs, such as the toad bladder, kidney and skin. We believe that the toad skin is composed of multiple layers and complex epidermal cells, and various components in skin secretions play an important role in the biological functions of the skin. This is reflected in our manuscript discussion and previous research findings.

2. Since toad *Bombina maxima* is not a typical model species, it is difficult to use gene editing methods to knock out $\beta\gamma$ -CAT, so we can only use multiple methods to carry out corresponding verification at the animal level. We believe that these attempts can be universally accepted and understood.

General comments:

Specific comments, with recommendations for addressing each comment.

1. Page 5, line 4-5. The two references mentioned are highly relevant. However recently, based upon considerable experimental evidence it has been concluded that amphibian skin plays a dual role in amphibian osmoregulation, see

Larsen EH (2021) Dual skin functions in amphibian osmoregulation. *Comparative Biochemistry and Physiology A. Molecular and Integrative Physiology* 253: 110869

It is the thesis of this review (and of previous recent studies by this author's laboratory)

that water balance of amphibians on land depends on integrated functions of the heterocellular surface epithelium, the subepidermal exocrine glands, and the cutaneous surface liquid.

I urge the authors to consider the above conceptual framework when discussing the function of skin in amphibian osmoregulation.

Response:

Thanks for your valuable suggestion. We quite agree with the latest ideas put forward in this review article. We combined the latest thesis with our experimental results to improve the discussion section of our manuscript. This makes our manuscript more concise and reasonable. Please see line 117, page 6.

General comments:

2. The authors' multi-experimental approaches are mostly based upon cellular and biochemical methods, which are handled professionally. This has resulted in a number of solid observations of interest for their own sake. However, it is also clear to me that taken together several individual results do not provide interrelated interpretations of consistent significance for the understanding of the macroscopic osmotic function of the skin. Rather than accepting this openly, the authors tend to speculate on their physiological importance. Serving as typical example, on page 15 line 450-457 the authors write:

“Obviously, these SELC proteins should be essential to cells in those cases that classic membrane solute carriers are blockaded or even absent. Our findings will serve as clues for uncovering the physiological roles of PFPs in other living organisms. Specifically, proliferating embryonic cells of oviparous animals depend entirely on catabolism of extracellular macromolecules deposited in the egg white and yolk. If macropinocytosis-driven by a PFP plays a role in oviparous animal embryonic development is an interesting subject for further investigation.”

Rather than merely asking the authors to clean their manuscript from this type of speculations remote from their actual experimental observations, I suggest that they make an effort to reconsider their general experimental protocol, which is a comparison between effects of hyperosmotic and isoosmotic exposures of the skin, respectively. More specifically, I suggest that the authors take into account that on land the skin is exposed to slightly hypoosmotic environment in terms of the cutaneous surface liquid. Exposure of the toad to external isoosmotic conditions might well simulate the cutaneous surface liquid, so that the experimental results obtained thus obtained can be as is. However, the other physiological challenge of the skin is experienced when the toad is in freshwater. This is a strongly hypoosmotic, not a hyperosmotic exposure. In other words, I ask for new experiments allowing for comparisons between these environmental conditions. I am aware that one of the incentives for this study is the observation that the ill-defined 'deletion

of toad skin secretions', in which $\beta\gamma$ -CAT is a major component, increased animal mortality under hypertonic stress. But taking the fact that few amphibian species (e.g., the green toad *B. viridis*) tolerate exposure to hypertonic saline, I find the above observation to be a curiosity rather than a response of importance for a general description of the toad's survival strategy.

Response:

Thanks for your careful review and valuable suggestion. Here we provide a brief summary:

1. Based on the latest experimental results, the discussion section of the manuscript was modified more rationally to better reflect our results and research significance.

2. We also agree with your suggestion about the convenience of animal experiments on manuscripts. And we did model the characteristics of toads in different osmotic environments. For example, the hypotonic treatment was modeled as hypertonic survival for 2 hours followed by reexposure to isotonic Ringer's solutions for 2 hours. This modeling method not only simulated the state of toad facing different osmotic environments, but also simulated the changes of related molecular characteristics during the transition between different osmotic states.

3. Although few toads can adapt to hypertonic conditions through their skin, their kidneys and bladders process hypertonic urine all the time. For land-adapted toads, the kidneys and bladder are also important organs for water balance, especially how to reuse water in urine. The biological functions of $\beta\gamma$ -CAT are explained in the figure 2 to 5 of our manuscript with bladder cells and kidney cells. We believe that this further demonstrates the rationality of CAT as a protein expressed in the multi-osmotic organ of toad *Bombina maxima*. And studies in these organs also help us to study the function of $\beta\gamma$ -CAT, because after all, as you said, there are so many molecules in the toad skin that are hard to ignore.

Figure:

References:

1. Challa, D. K. *et al.* Neonatal Fc receptor expression in macrophages is indispensable for IgG homeostasis. *mAbs* **11**, 848–860 (2019).
2. Lencer, W. I. & Blumberg, R. S. A passionate kiss, then run: exocytosis and recycling of IgG by FcRn. *Trends Cell Biol.* **15**, 5–9 (2005).
3. Suzuki, M., Hasegawa, T., Ogushi, Y. & Tanaka, S. Amphibian aquaporins and adaptation to terrestrial environments: A review. *Comp. Biochem. Physiol. A. Mol. Integr. Physiol.* **148**, 72–81 (2007).
4. Takata, K. Aquaporins: water channel proteins of the cell membrane. *Prog. Histochem. Cytochem.* **39**, 1–83 (2004).
5. McManus, M. L., Churchwell, K. B. & Strange, K. Regulation of cell volume in health and disease. *N. Engl. J. Med.* **333**, 1260–1267 (1995).

Reviewers' comments:

Reviewer #1 (Remarks to the Author):

In the revised manuscript, Zhao et al. have satisfactorily addressed most of my concerns. Two issues remain with regard the section on macropinocytosis.

1. Original comment #1 concerning the increase of macropinocytosis by IgG treatment (sometimes to a dramatic extent): I understand the author's reasoning, and as they point out other approaches support the conclusion that bg-CAT promotes macropinocytosis. However, this is a relevant issue that is critical for the reader to adequately interpret the data. The authors should be very open about this and discuss this caveat of the experiments with neutralizing antibodies explicitly in the results section.

2. Original comment #3: The revised microscopy data in Figure S3 remain unconvincing. In toad cells (S3e, f) bg-CAT staining is observed in large parts of the cell and it is utterly unclear whether a) bg-CAT localizes to macropinocytic vesicles and b) whether bg-CAT displays any specific co-localization with dextran. The authors argue that experiments with toad cells in culture are technically challenging. If this prevents a clear demonstration of the colocalization of bg-CAT and dextran, then the results should be removed from the manuscript.

In MDCK cells (S3g), dextran still seems to display an entirely nuclear localization, and it is very hard to imagine that this signal localizes to macropinosomes. Again, if the experiments cannot be performed in a more convincing way, the figure should be removed.

Reviewer #2 (Remarks to the Author):

Having studied the revised manuscript dealing with a classical physiological problem, I have to conclude that I still miss to see a well-defined quantitative role for the pinocytotic poreforming protein, $\beta\gamma$ -CAT, in toads' osmoregulation; neither under hyperosmotic conditions nor at any other type of environmental osmotic challenge.

Peer Review File

Manuscript Title: A pore-forming protein drives macropinocytosis to facilitate toad water maintaining

Your manuscript entitled "A pore-forming protein drives macropinocytosis to facilitate toad water maintaining" has now been seen again by 2 referees. We are interested in the possibility of publishing your study in Communications Biology, but would like to consider your response to these concerns in the form of a revised manuscript before we make a final decision on publication.

We therefore invite you to revise and resubmit your manuscript, taking into account the points raised. In particular, there are comments remaining from reviewer 1 that we suggest further exploration of, with experimental work if necessary. We ask that only textual revisions be made to address reviewer 2's comments about the potential limitations they state.

Please highlight all changes in the manuscript text file.

We are committed to providing a fair and constructive peer-review process. Do not hesitate to contact us if you wish to discuss the revision in more detail or if there are specific requests from the reviewers that you believe are technically impossible or unlikely to yield a meaningful outcome.

Response:

Thank you very much for your positive and valuable comments on our research work. We greatly appreciate your consideration of our manuscript. We have revised our original manuscript according to the suggestions of the reviewers. We hope this version of the manuscript will give you satisfaction. We have used a yellow background in the manuscripts to highlight all the changes.

Reviewer Comments & Author Response

The following is a point-by-point response to the reviewers' comments:

Reviewers' comments:

Reviewer #1:

In the revised manuscript, Zhao et al. have satisfactorily addressed most of my concerns. Two issues remain with regard the section on macropinocytosis.

1. Original comment #1 concerning the increase of macropinocytosis by IgG treatment (sometimes to a dramatic extent): I understand the author's reasoning, and as they point out other approaches support the conclusion that bg-CAT promotes macropinocytosis. However, this is a relevant issue that is critical for the reader to adequately interpret the data. The authors should be very open about this and discuss this caveat of the experiments with neutralizing antibodies explicitly in the results section.

Response:

We thank the reviewers for the great comments and we have now clearly discussed the point in the results and discussion sections. We have used a yellow background in the manuscripts to highlight all the changes.

First, two key experimental results were added to the supplemental information to demonstrate the potential of rabbit IgG to stimulate pinocytosis. One was that the IgG receptor were present on toad cells (Fig. S3c) and the ability of control rabbit antibody and rabbit-derived $\beta\gamma$ -CAT antibody to stimulate pinocytosis was equally effective on MDCK (Fig. S3d). Secondly, we given a reasonable description of this new result in the results and discussion.

Please see line 244-253, 256, page 10; line 406-408, page 16 in the new manuscript.

Please see line 80-81, page 3; line 150-152, page 6; line 357-358, page 18; line 309-317, page 13 and Fig. S3c and d in the new supplemental information.

2. Original comment #3: The revised microscopy data in Figure S3 remain unconvincing. In toad cells (S3e, f) bg-CAT staining is observed in large parts of the cell and it is utterly unclear whether a) bg-CAT localizes to macropinocytic vesicles and b) whether bg-CAT displays any specific co-localization with dextran. The authors argue that experiments with toad cells in culture are technically challenging. If this prevents a clear demonstration of the colocalization of bg-CAT and dextran, then the results should be removed from the

manuscript.

In MDCK cells (S3g), dextran still seems to display an entirely nuclear localization, and it is very hard to imagine that this signal localizes to macropinosomes. Again, if the experiments cannot be performed in a more convincing way, the figure should be removed.

Response:

Yes, we've recognized that problem now. We think this experiment is important for our central point, so we updated the experimental method to reverify this result. Positive results have been obtained and updated. Specifically, we labeled $\beta\gamma$ -CAT with RBITC, which eliminates the need for the complex steps of immunofluorescence in the later stage to affect experimental results. Our new experimental evidence suggests that $\beta\gamma$ -CAT can increase the macropinocytosis of MDCK cells to 70 kDa dextran, and the two have cytoplasmic co-localization (Fig. S3g). In addition, we did not get better results with toad cells, so we deleted this data to avoid unknown effects.

Please see line 244-253, 256-257, page 10; line 406-408, page 16 in the new manuscript.

Please see line 83-94, 97-100, 108-111, page 4; line 112-113, 115-116, page 5; line 320-323, page 16 and Fig. S3g in the new supplemental information.

Reviewer #2 (Remarks to the Author):

Having studied the revised manuscript dealing with a classical physiological problem, I have to conclude that I still miss to see a well-defined quantitative role for the pinocytotic poreforming protein, $\beta\gamma$ -CAT, in toads' osmoregulation; neither under hyperosmotic conditions nor at any other type of environmental osmotic challenge.

Response:

Thank you for your concerns. We think that $\beta\gamma$ -CAT stimulated and participated in cell macropinocytosis and exosome release, promoting water and Na^+ uptake and regulating AQP localization. We believed that the transport of water and Na^+ is a good indicator of osmotic regulation from a functional point of view. Furthermore, in order to increase the rationality of the article and avoid unnecessary misunderstanding, we have added relevant rationality elaboration in the discussion.

Please see line 387-389, page 16; line 473-477, page 19 in the new manuscript.

Other Response:

We have updated the mailing addresses of some authors.

REVIEWERS' COMMENTS:

Reviewer #1 (Remarks to the Author):

The authors have satisfactorily addressed my concerns, and I recommend the manuscript for publication in Communications Biology.